# A meta-analysis of the association between inflammatory cytokine polymorphism and neonatal sepsis

**Jiaojiao Liang**[‡], **Yan Su**[‡], **Na Wang, Xiaoyan Wang, Ling Hao**[*], **Changjun Ren**[*]

The First Hospital of Hebei Medical University, Shijiazhuang, Hebei, China

‡ JL and YS are equally to this work as first coauthors.
* doctorhao@hebmu.edu.cn (LH); renchangjun@hebmu.edu.cn (CR)

**Data Availability Statement:** All relevant data are within the manuscript and its Supporting Information files.

## Abstract

### Objective

The purpose of this study is to investigate the relationship between single nucleotide polymorphisms of inflammatory cytokines and neonatal sepsis through meta-analysis.

### Methods

We collected research literature on the correlation between inflammatory cytokine polymorphisms and neonatal sepsis published before August 2023 through computer searches of databases such as PubMed, Embase, etc. The Stata 14.0 software was utilized for Meta-analysis. To assess heterogeneity, the chi-squared Q-test and $I^2$ statistics were used. The Egger and Begg tests were conducted to determine the possibility of publication bias.

### Results

After reviewing 1129 articles, 29 relevant articles involving 3348 cases and 5183 controls were included in the study. The meta-analysis conducted on IL-1βrs1143643 polymorphism revealed significant findings: the T allele genotype has a lower risk of neonatal sepsis(P = 0.000, OR = 0.224, 95% CI: 0.168–0.299), while the TC and TT genotypes showed an increased risk(TC: P = 0.000, OR = 4.251, 95% CI: 2.226–8.119; TT: P = 0.019, OR = 2.020, 95% CI: 1.122–3.639). Similarly, newborns with the IL-6-174 CC genotype had a significantly higher risk of sepsis(P = 0.000, OR = 1.591, 95% CI: 1.154–2.194), while those with the IL-8-rs4073 TT (P = 0.003, OR = 0.467, 95% CI: 0.280–0.777)and TT + AA(P = 0.003, OR = 0.497, 95% CI: 0.315–0.785) genotypes had a significantly lower risk of sepsis. For the IL-10-1082 gene, newborns with the AA genotype(P = 0.002, OR = 1.702, 95% CI: 1.218–2.377), as well as those with the AA + GA genotype(P = 0.016, OR = 1.731, 95% CI: 1.108–2.705), had a significantly higher risk of sepsis. Lastly, newborns carrying the TNF-α–308 A allele (P = 0.016, OR = 1.257, 95% CI: 1.044–1.513)or the AA genotype(P = 0.009, OR = 1.913, 95% CI: 1.179–3.10) have a significantly increased risk of sepsis. Notwithstanding, additional studies must be included for validation. Applying these cytokines in clinical practice and integrating them into auxiliary examinations facilitates the early detection of

**Funding:** The author(s) received no specific funding for this work.

**Competing interests:** The authors have declared that no competing interests exist.

susceptible populations for neonatal sepsis, thereby providing a new diagnostic and therapeutic approach for neonatal sepsis.

## 1. Introduction

Sepsis is an infectious disease wherein pathogenic bacteria invade the body and release toxins that cause systemic inflammatory reactions [1]. Neonates are extremely prone to sepsis due to high-risk factors such as the infant immune system, changes in the postnatal environment, and maternal infectious diseases [2, 3]. Septicemia is an important factor that affects the survival rate of newborns, and its incidence rate and mortality are very high [4–8]. The clinical manifestations of sepsis are similar to those of many diseases, and the relevant auxiliary investigations lack specificity for its diagnosis. This makes it difficult to identify susceptible populations of sepsis in clinical practice, thus impeding timely diagnosis and treatment.

Currently, the gold standard for diagnosing sepsis remains blood culture; however, this requires a long time and is prone to false-negative results, making it less effective in the early diagnosis of sepsis [9–11]. Although there are various biomarkers for neonatal sepsis, including classical, non-traditional, and novel types [12], each of them has its own limitations. Among classic biomarkers, a decrease in peripheral blood leukocyte count has a higher specificity but lower sensitivity in diagnosing neonatal sepsis [12], a decrease in neutrophil count is more sensitive to diagnosing neonatal sepsis but is more susceptible to factors such as delivery route and gestational age [13], a decrease in platelet count alone has insufficient sensitivity and specificity in diagnosing neonatal sepsis [14], C-reactive protein (CRP) and procalcitonin (PCT) it is of great significance in the diagnosis of bacterial infectious diseases [15]. The combination of the two can be used to diagnose neonatal sepsis [16];however, it is easily influenced by non-infectious diseases (such as trauma). Nontraditional biomarkers are mainly cytokines, which play a crucial role in the host immune response, immune regulation, and immune-inflammatory response [17, 18]. Interleukin-1 (IL-1) has a significant effect on systemic infections. Interleukin-6 (IL-6) is a key regulatory factor that is highly sensitive to early bacterial infections in newborns [12]. Interleukin-8 (IL-8) has a regulatory effect on both acute and chronic inflammatory responses and has a chemotactic effect on neutrophils. It can detect early onset sepsis (EOS). Neonatal sepsis may be more sensitive to the combination of IL-8 and CRP, guiding clinicians in the rational use of antibiotics [19]. Interleukin-10 (IL-10) has anti-inflammatory and immunomodulatory effects and plays a significant regulatory role in the occurrence, development, and outcome of neonatal sepsis [20]. Tumor necrosis factor-$\alpha$ (TNF-$\alpha$) significantly impacts severe infections, with its increase in serum concentration differs it from CRP and PCT and is not affected by gestational age and postnatal age [17]. In bacterial infectious diseases, such as septicemia, the increase of serum concentration of IL-1, IL-6, IL-8, IL-10, and TNF-$\alpha$ is generally earlier than the clinical manifestation of the newborn [19]. This increase can help detect neonatal septicemia early and provide timely intervention measures. However, there is currently no unified diagnostic threshold in clinical practice, and the increase in cytokine levels may be affected by multiple factors, such as the mother's infectious disease and transient increase after environmental stimulation, thus, leading to false-positive results. New biomarkers, such as the leukocyte surface antigen CD64, are not expressed under normal physiological conditions [21] but are significantly elevated when sepsis occurs in the body [22]. However, some scholars [23] have found that the specificity and sensitivity of CD64 are insufficient, and its diagnostic role in neonatal sepsis is not ideal. In summary, these biomarkers have shortcomings that make early diagnosis of neonatal sepsis difficult.

With the completion of the Human Genome Project and our deeper understanding of molecular genetics, an increasing number of researchers have discovered [24] that genetic variations at gene loci have a significant impact on the interactions between high-risk factors for sepsis. In recent years, epidemiological studies have indicated [25] that genetic polymorphisms encoding inflammatory cytokines may affect the occurrence, development, and outcomes of sepsis [26, 27]. Although many researchers have used cytokine gene polymorphisms to study the genetic susceptibility to neonatal sepsis, the results vary. Therefore, this study searched relevant literature from various databases and used meta-analysis to analyze the cytokines (including IL-1, IL-6, IL-8, IL-10, and TNF-α), aiming to provide more reliable evidence-based medicine for the pathogenesis, pathophysiology, early diagnosis and treatment, and prognosis of neonatal sepsis.

## 2. Materials and methods

### 2.1 Document retrieval

Various databases, including the China National Knowledge Infrastructure, WanFang Data, China Biological Medicine Disc, PubMed, Embase, Cochrane Library, and Web of Science, were searched to compile literature on the connection between inflammatory cytokine polymorphisms and neonatal sepsis. The search was conducted from the inception of the database construction until August 2023. Based on similar literature [28], we devised search terms as "Infant, Newborn", "Infant, Premature", "Infant,Low-Birth-Weight", "Infant,Very Low Birth Weight", "Sepsis", "Bacteremia", "Hemorrhagic Septicemia", "Cytokines", "Interleukins", "Tumor Necrosis Factor-alpha", "Polymorphism, Genetic." We combined medically relevant keywords with broader terms to ensure a comprehensive search for relevant articles. In instances of duplicate publications, we selected those with the most comprehensive data for inclusion in the study. We scrutinized repetitive publications, comparing their content, and randomly included studies if the content is completely repetitive. In cases of content disparity, a detailed comparison was conducted with preference given to studies providing the most comprehensive data. If the differences persisted, both or more studies were included.

### 2.2 Inclusion and exclusion criteria

**Inclusion criteria:** 1. Relevant studies on cytokine polymorphisms and neonatal sepsis published in China and abroad have used case-control studies or cohort designs; 2. Definitive diagnosis of neonatal sepsis; 3. The research subjects of the literature were newborns; 4. The control group was the non-sepsis group (patients with suspected sepsis but no sepsis or healthy newborns); 5. The literature has sufficient data to extract genotype distribution data for cases and controls; 6. The reported genotype distribution was in Hardy Weinberg equilibrium.

**Exclusion criteria:** 1.The subjects were not newborns; 2. Experimental data could not be obtained or were insufficient; 3. The research content of the literature is repeated; 4. Meta-analysis, review, or systematic review; 5. Only allele carriers were included in this study.

### 2.3 Literature screening

Two investigators independently reviewed and evaluated each study, and in cases of disagreement, an additional researcher evaluated the literature. Duplicate content references were eliminated first, followed by references unrelated to the research content. The remaining documents were scrutinized against the inclusion criteria with ineligible studies excluded. Any uncertainties or missing data were addressed by contacting the corresponding authors of the article. The literature screening process is shown in Figs 1–5.

## 2.4 Data extraction

From the 29 included studies, we extracted information including the first author, year of publication, country, number of case and control groups, the origin of the control group, and genotype composition in the case and control groups.

## 2.5 Quality evaluation

Being an observational study that included case-control and cohort studies, we employed the use of the Newcastle-Ottawa Scale (NOS), suitable for evaluating case-control and cohort studies [29], to evaluate the quality of the 29 articles included in the study. The NOS scale evaluated of the selection method for the case and control groups, the assessment method of exposure factors, and the comparability of the case and control groups. The included studies were comprehensively evaluated based on these three aforementioned sections. The higher the number of stars obtained after the evaluation, the better the quality of the literature. The best one was 10 stars, and generally, the literature with ≥5 stars could be included in our study.

## 2.6 Statistical analysis

In the final 29 literatures included, data were extracted according to the content of the data extraction table mentioned above and imported into Stata14.0 software for statistical analysis, with a significant level set at α = 0.05. Given this study's enature as a case-control genetic correlation study, a Hardy Weinberg (HWE) test was conducted on the control group to determine if the samples are from the same Mendelian population in order to reduce sampling bias in case-control studies [30]. However, the effectiveness of the HWE test is not high [31]. We reported the *P*-value of the HWE test results. With *P*>0.05 indicating conformity to the law of genetic balance. Studies that did not comply with HWE were subjected to a combined analysis, followed by a sensitivity analysis to assess the stability of the results. The relationship between cytokine polymorphismas and neonatal sepsis was evaluated using the effect variable odds ratio(OR) and 95% confidence interval (CI). Heterogeneity assessment was conducted through the chi-square Cochran's Q test and $I^2$ statistics, selecting a fixed effects model when *P*>0.1 and $I^2$≤50%. Otherwise, a random-effects model was chosen. Sensitivity and subgroup analyses was performed to identify sources of heterogeneity in studies with significant heterogeneity. Begg's and Egger's tests were used to detect publication biases in the included studies, with *P*>0.05, indicating no publication bias. TSA software was used to conduct a sequential analysis of the included studies, obtain the sample size required for meta-analysis, and verify whether the results of the included studies were reasonable. Multiple tests were performed using the Bonferroni correction. The false-positive result reporting rate (FPRP) was used to determine false-positive research results.

# 3. Results

## 3.1 General situation and quality of inclusion in the study

We initially searched 1129 articles, of which 202 were in Chinese and 927 were in English. After a series of screening, 29 articles [32–52] were finally included in the meta-analysis, including 4 articles [32–35] related to IL-1, 9 articles [34, 36–43] to IL-6, 2 articles [36, 44] related to IL-8, 6 articles [38, 43, 44–47] related to IL-10, and 8 articles [34, 36, 41, 48–52] related to TNF-α (see Figs 1–5). Basic information, genotype distribution, and quality of the included studies (NOS scores ≥ 5 for all included studies) are shown in Table 1. The total sample size across the literature was 8531. The case group comprised 3348 newborns with sepsis or suspected sepsis, while the control group consisted of 5183 healthy or non-sepsis newborns

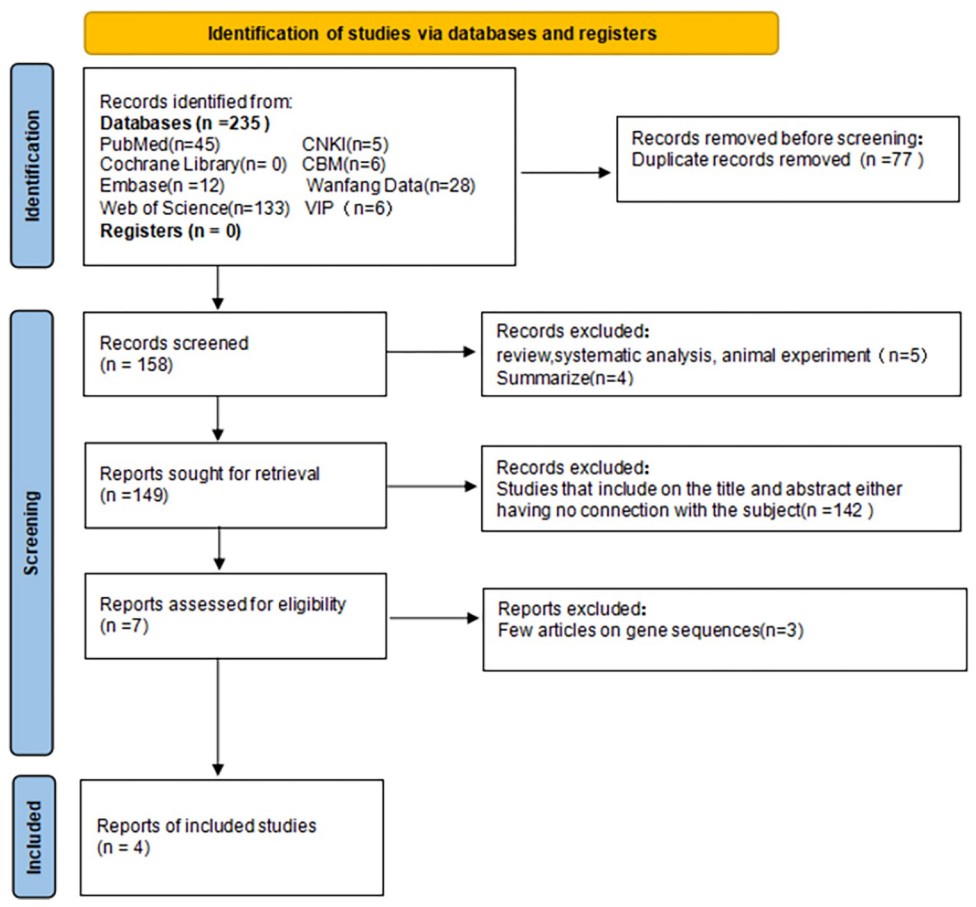

**Fig 1. Flow chart of IL-1 literature screening.**

without sepsis. Among the four studies on IL-1, there were 468 cases in the case group and 248 cases in the control group. Three studies were conducted among Asians and one study among Europeans. Of the nine studies on IL-6, 882 cases were in the case group while 1240 cases were in the control group. Two studies were conducted among Asians, six among Europeans, and one among Africans. Two studies on IL-8 involved 166 cases in the case group and 183 cases in the control group, with one each among Asians and Europeans. Among the six studies on IL-10, there were 494 cases in the case group and 401 cases in the control group, with three studies conducted among Asians, two among Europeans, and one among Africans. Regarding TNF- α, among the eight studies, Härtel C. [48]. included two cohorts in one article, which were considered independent studies in this study (Härtel C. and Härtel C. [48]). There were 1340 cases in the case group and 3111 cases in the control group, with three studies conducted among Asians and six studies conducted among Europeans.

## 3.2 Hardy Weinberg (HWE) test results (see Table 1)

Due to genotyping errors, selection bias, and inappropriate stratification, HWE bias could occur [31]. Therefore, before summarizing and analyzing the data, HWE tests were performed for each control group and we used Stata14.0 software to calculate the HW-$P$ value and the adjusted HW-$P$ value. The results are summarized in Table 1. Except for a few studies [32, 33, 37, 43, 50], $P$ was greater than 0.05, indicating that most studies complied with the law of

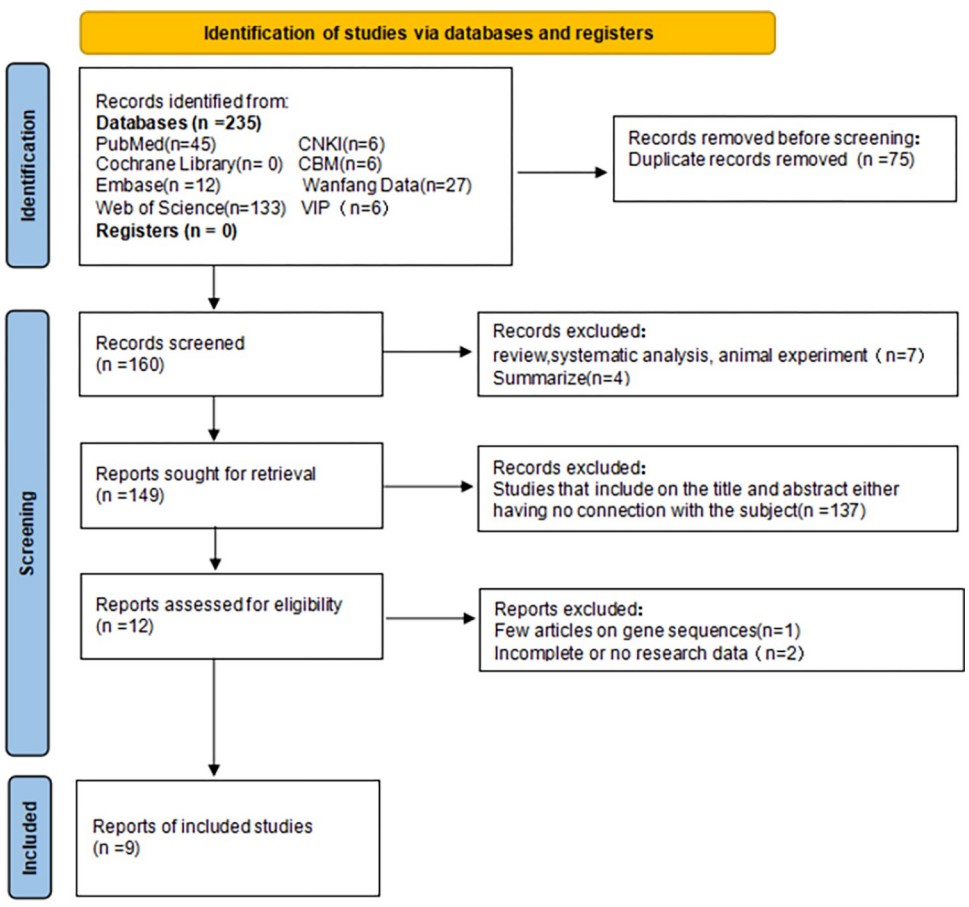

**Fig 2. Flow chart of IL-6 literature screening.**

genetic balance. We did not exclude studies that deviated from the HWE law, but instead conducted a meta-analysis along with other studies, followed by a sensitivity analysis. Ultimately, We found that only one study on IL-1 that deviated from the HWE law had a significant impact on the results under the allele model, and it was excluded. Other studies that deviate from the HWE law had a relatively low impact on the results.

### 3.3 Selection of genetic models

This article presents a study on genetic correlation, which requires separate analyses of multiple gene models to achieve the goal of multiple pairwise comparisons of these genotypes and reduce the occurrence of type I errors. The commonly used gene models currently include the allele model (A vs. B), dominant gene model (AA+AB vs. BB), recessive gene model (AA vs. AB+BB), heterozygous gene model (AB vs. BB), homozygous gene model (AA vs. BB), and additive gene model (AA+BB vs. AB) [53].

### 3.4 Heterogeneity assessment (see Table 2)

In our study, four studies related to IL-1βrs1143643 showed high heterogeneity among the above six models (allele model: $I^2$ = 92.9%, $P$ = 0.000; dominant gene model: $I^2$ = 92.4%, $P$ = 0.000; recessive gene model: $I^2$ = 89.9%, $P$ = 0.000; heterozygous gene model: $I^2$ = 87.4%,

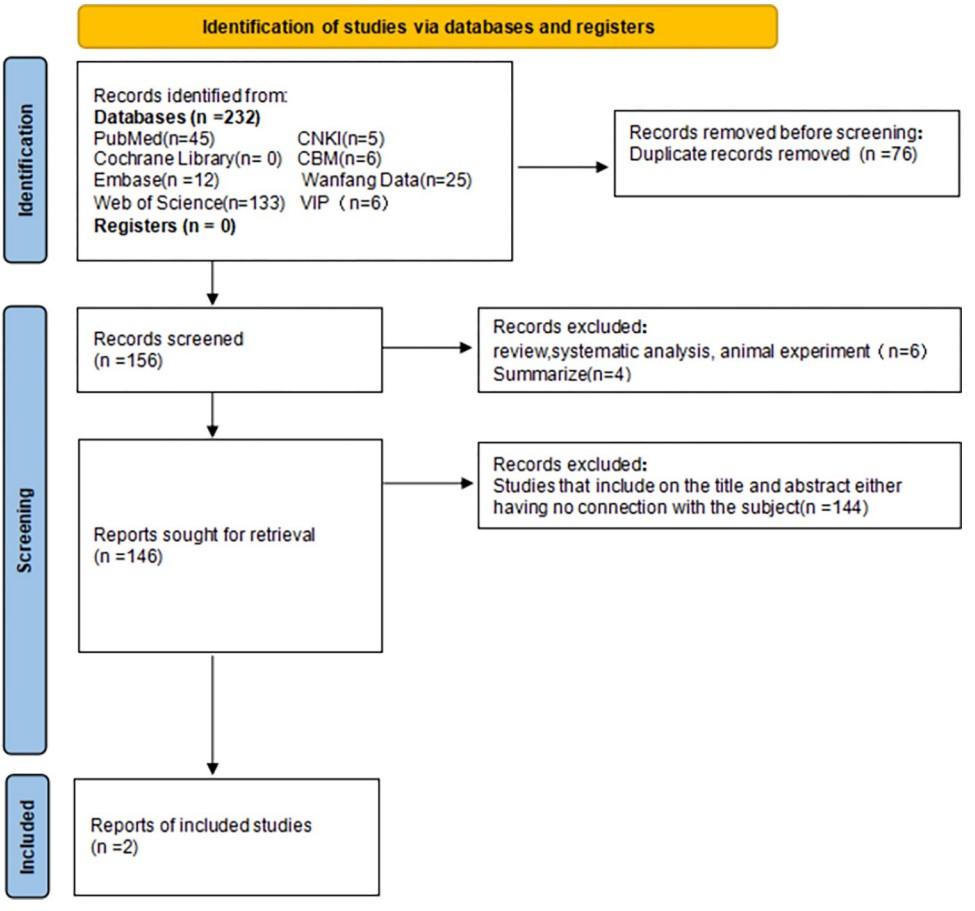

**Fig 3. Flow chart of IL-8 literature screening.**

$P = 0.000$; homozygous gene model: $I^2 = 92.3\%$, $P = 0.000$; additive gene model: $I^2 = 85.0\%$, $P = 0.000$).

For nine studies involving IL-6-174G/C, heterogeneity was high in dominant gene model, heterozygous gene model and additive gene model (dominant gene model: $I^2 = 64.9\%$, $P = 0.004$; heterozygous gene model: $I^2 = 72.1\%$, $P = 0.001$; additive gene model: $I^2 = 75.4\%$, $P = 0.000$), there was no heterogeneity in the other models (allele model: $I^2 = 39.0\%$, $P = 0.119$; recessive gene model: $I^2 = 0.00\%$, $P = 0.434$; homozygous gene model: $I^2 = 0.00\%$, $P = 0.686$).

Two studies on IL-8-rs4073 showed no heterogeneity among the six gene models (allele model: $I^2 = 0.00\%$, $P = 0.566$; dominant gene model: $I^2 = 0.00\%$, $P = 0.740$; recessive gene model: $I^2 = 0.00\%$, $P = 0.571$; heterozygous gene model: $I^2 = 0.00\%$, $P = 0.938$; homozygous gene model: $I^2 = 0.00\%$, $P = 0.604$; additive gene model: $I^2 = 0.00\%$, $P = 0.749$).

In six studies involving IL-10-1082 G/A, there was no heterogeneity in the heterozygous and additive gene models (heterozygous gene model: $I^2 = 40.9\%$, $P = 0.149$; additive gene model: $I^2 = 0.00\%$, $P = 0.760$). Heterogeneity was evident in the remaining gene models (allele model: $I^2 = 75.6\%$, $P = 0.001$; dominant gene model: $I^2 = 69.9\%$, $P = 0.010$; recessive gene model: $I^2 = 55.9\%$, $P = 0.045$; homozygous gene model: $I^2 = 72.7\%$, $P = 0.005$).

In the eight studies associated with TNF-α-308g /A(rs1800629), there was no heterogeneity in the homozygous gene model ($I^2 = 41.2\%$, $P = 0.131$). In the other gene models, significant heterogeneity was observed(allele model: $I^2 = 66.2\%$, $P = 0.004$; dominant gene model: $I^2 =$

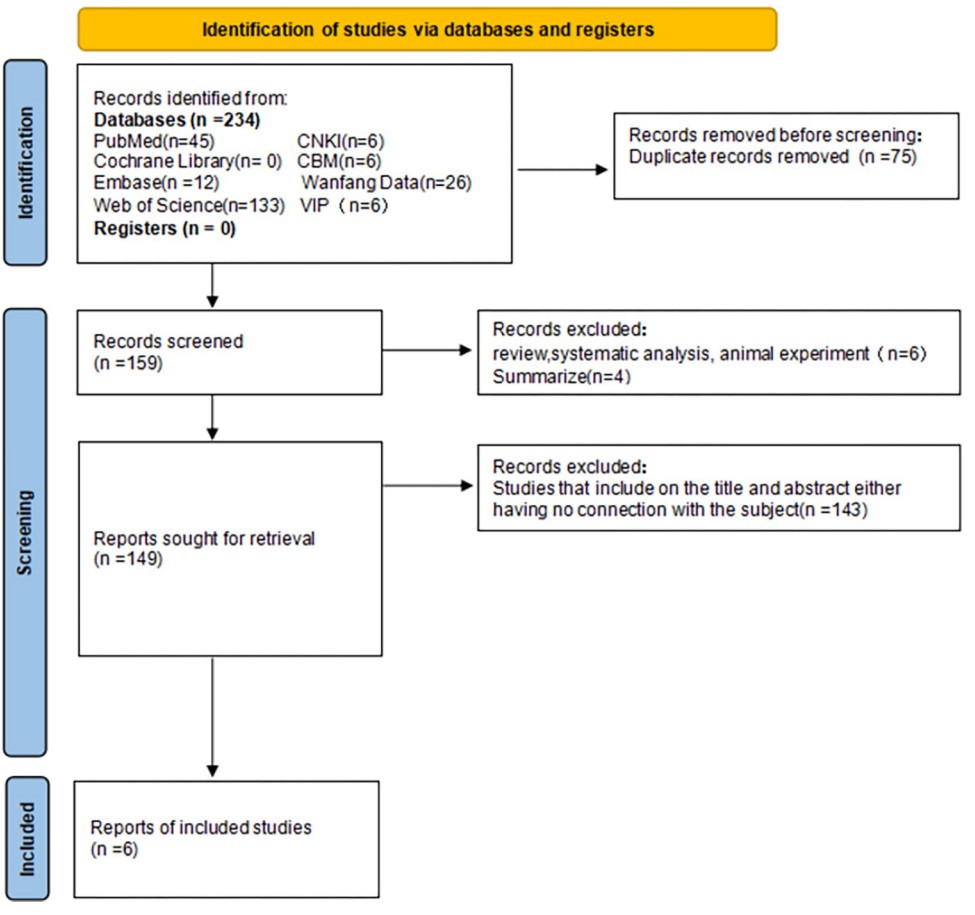

**Fig 4. Flow chart of IL-10 literature screening.**

54.7%, $P$ = 0.024; recessive gene model: $I^2$ = 52.7%, $P$ = 0.061; heterozygous gene model: $I^2$ = 50.8%, $P$ = 0.047; additive gene model: $I^2$ = 50.1%, $P$ = 0.051).

## 3.5 The relationship between polymorphisms of various cytokines and neonatal sepsis (see Table 2)

The results show that the OR value of IL-1βrs1143643 polymorphism merged in the allele model was statistically significant ($P$ = 0.000), indicating that the T allele was associated with a lower risk of sepsis in newborns (OR = 0.224, 95% CI: 0.168–0.299). The OR value merged in the heterozygous gene model was also statistically significant ($P$ = 0.000), suggesting an increase the risk of neonatal sepsis with the TC genotype (OR = 4.251, 95% CI: 2.226–8.119). Similarly, the OR value within the homozygous gene model was statistically significant ($P$ = 0.019), indicating an increase risk of sepsis in newborns with the TT genotype (OR = 2.020, 95% CI: 1.122–3.639). There was no statistically significant difference between the other gene models (dominant gene model: $P$ = 0.059, OR = 2.785,95% CI: 0.963–8.052; recessive gene model: P = 0.089, OR = 1.567,95% CI: 0.934–2.631; additive gene model: $P$ = 0.822, OR = 0.904,95% CI: 0.377–2.171).

The results showed that the OR value of the IL-6-174G/C polymorphism merged in the recessive gene model was statistically significant ($P$ = 0.000), indicating a significant increase in the risk of neonatal sepsis with the CC genotype (OR = 1.591, 95% CI: 1.154–2.194).

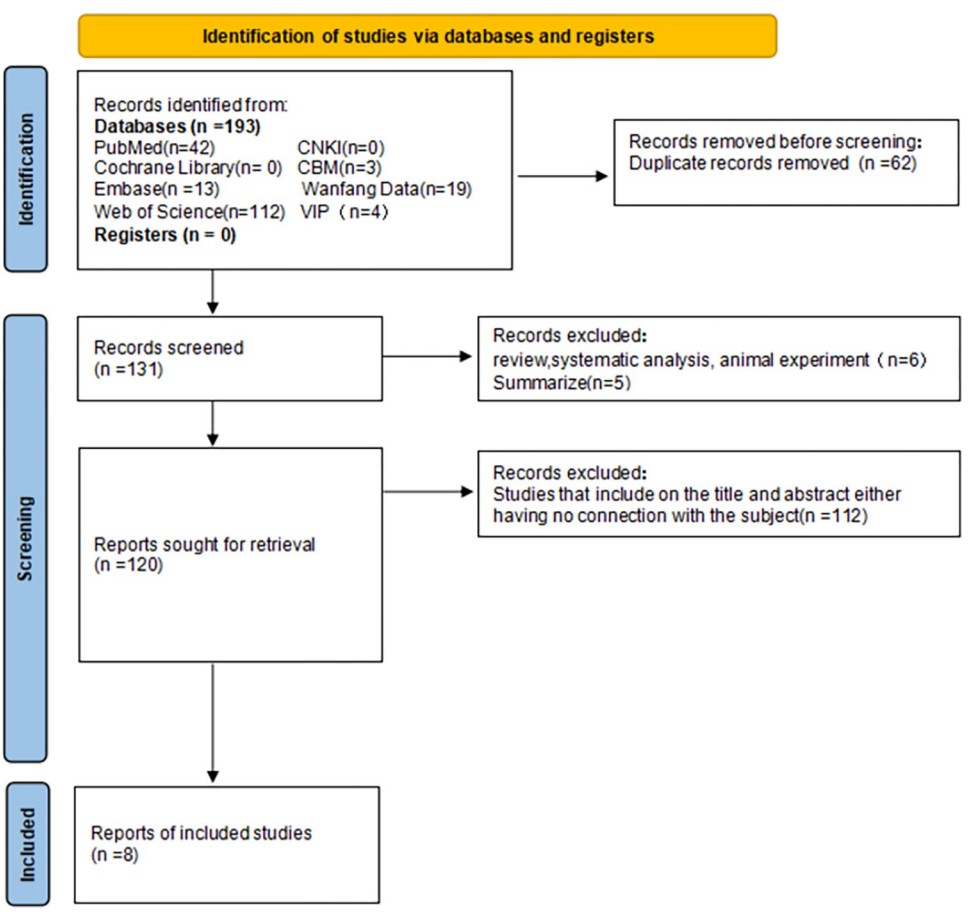

**Fig 5. Flow chart of TNF-α literature screening.**

Additionally, the OR value merged in the homozygous gene model was statistically significant ($P$ = 0.034), indicating a significant increase in the risk of sepsis in newborns with the CC genotype (OR = 1.471, 95% CI: 1.030–2.101). However, there were no statistically significant differences in the other gene models (allele model: $P$ = 0.101, OR = 1.140, 95% CI: 0.974–1.335; dominant gene model: $P$ = 0.950, OR = 1.012, 95% CI: 0.689–1.488; heterozygous gene model: $P$ = 0.916, OR = 1.021, 95% CI: 0.692–1.508; additive gene model: $P$ = 0.635, OR = 1.059, 95% CI: 0.835–1.343).

The results showed that the OR value of the IL-8-rs4073 polymorphism merged in the recessive gene model was statistically significant ($P$ = 0.003), indicating a significant reduction in the risk of sepsis in newborns with the TT genotype (OR = 0.467, 95% CI: 0.280–0.777). The OR value merged in the additive gene model was statistically significant ($P$ = 0.003), indicating a significant reduction in the risk of sepsis in newborns with the TT+AA genotypes (OR = 0.497, 95% CI: 0.315–0.785). There was no statistically significant difference in the other gene models (allele model: $P$ = 0.087, OR = 0.756, 95% CI: 0.549–1.041; dominant gene model: $P$ = 0.713, OR = 1.112, 95% CI: 0.633–1.951; heterozygous gene model: $P$ = 0.149, OR = 1.548, 95% CI: 0.855–2.806; homozygous gene model: $P$ = 0.187, OR = 0.639, 95% CI: 0.328–1.234).

Moreover, the results showed that the OR value of the IL-10-1082 G/A polymorphism merged in the dominant gene model was statistically significant ($P$ = 0.016), indicating a

**Table 1. Basic information of included literature.**

### IL-1βrs1143643

| Number | Include studies | Race | Genotyping methods | Quality evaluation | Case/Control | Case | | | Control | | | $P_{HWE}$ |
|---|---|---|---|---|---|---|---|---|---|---|---|---|
| | | | | | | CC | CT | TT | CC | CT | TT | |
| 1 | Mustarim Mustarim 2019 | Asia | PCR | 5 | 30/30 | 7 | 20 | 3 | 9 | 9 | 12 | 0.03 |
| 2 | Susanna Esposito 2014 | Europe | PCR | 5 | 199/100 | 75 | 86 | 32 | 52 | 33 | 10 | 0.18 |
| 3 | GAMAL ALLAM 2015 | Asia | ELISA | 6 | 137/68 | 62 | 58 | 17 | 3 | 32 | 33 | 0.16 |
| 4 | Zhao Xiaofen2020 | Asia | PCR | 7 | 100/50 | 48 | 28 | 24 | 17 | 27 | 6 | 0.41 |

### IL6-174G/C

| Number | Include studies | Race | Genotyping methods | Quality evaluation | Case/Control | Case | | | Control | | | $P_{HWE}$ |
|---|---|---|---|---|---|---|---|---|---|---|---|---|
| | | | | | | GG | GC | CC | GG | GC | CC | |
| 1 | Andra's Treszl 2003 | Europe | PCR-RFLP | 6 | 68/35 | 33 | 27 | 8 | 19 | 15 | 1 | 0.32 |
| 2 | PETER AHRENS [37] 2004 | Europe | PCR | 5 | 50/306 | 24 | 21 | 5 | 97 | 177 | 32 | 0.00 |
| 3 | R J o h n B a i e r [38] 2006 | Europe | PCR | 5 | 114/119 | 94 | 18 | 2 | 110 | 9 | 0 | 0.66 |
| 4 | W G o¨pel 2006 | Europe | PCR | 5 | 97/320 | 29 | 50 | 18 | 128 | 143 | 49 | 0.38 |
| 5 | GAMAL ALLAM 2015 | Asia | ELISA | 6 | 137/68 | 52 | 52 | 33 | 14 | 49 | 5 | 0.75 |
| 6 | NM Karakaş 2018 | Asia | PCR-RFL*P* | 6 | 60/123 | 47 | 13 | | 86 | 37 | | accord |
| 7 | Tatjana Varljen 2019 | Europe | PCR | 6 | 282/189 | 149 | 148 | 45 | 57 | 61 | 13 | 0.56 |
| 8 | Khalit S. Khaertynov 2016 | Europe | PCR | 5 | 20/10 | 3 | 13 | 4 | 3 | 5 | 2 | 0.97 |
| 9 | Hesham Abdel-Hady 2009 | Africa | PCR | 6 | 54/70 | 17 | 26 | 11 | 8 | 32 | 11 | 0.71 |

### IL8-rs4073

| Number | Include studies | Race | Genotyping methods | Quality evaluation | Case/Control | Case | | | Control | | | $P_{HWE}$ |
|---|---|---|---|---|---|---|---|---|---|---|---|---|
| | | | | | | AA | AT | TT | AA | AT | TT | |
| 1 | Andra´ s Treszl 2003 | Europe | PCR-RFLP | 5 | 66/133 | 15 | 37 | 16 | 10 | 20 | 5 | 0.02 |
| 2 | Zhao Xiaofen 2020 | Asia | PCR | 6 | 100/50 | 15 | 55 | 20 | 10 | 23 | 17 | 0.66 |

### IL-10-1082 G/A

| Number | Include studies | Race | Genotyping methods | Quality evaluation | Case/Control | Case | | | Control | | | $P_{HWE}$ |
|---|---|---|---|---|---|---|---|---|---|---|---|---|
| | | | | | | GG | GA | AA | GG | GA | AA | |
| 1 | Andra's Treszl 2003 | Europe | PCR-RFLP | 6 | 68/35 | 14 | 38 | 14 | 28 | 50 | 51 | 0.32 |
| 2 | R J o h n B a i e r [38] 2006 | Europe | PCR | 5 | 148/145 | 14 | 71 | 63 | 26 | 74 | 45 | 0.64 |
| 3 | Bao Jin 2006 | Asia | PCR | 5 | 54/40 | 33 | 18 | 3 | 13 | 19 | 9 | 0.56 |
| 4 | Hesham Abdel-Hady2009 | Africa | PCR | 6 | 54/70 | 6 | 37 | 11 | 9 | 47 | 14 | 0.00 |
| 5 | Yu Xin 2022 | Asia | PCR | 6 | 114/60 | 8 | 43 | 63 | 8 | 31 | 21 | 0.51 |
| 6 | Zhao Xiaofen2023 | Asia | PCR | 6 | 56/51 | 0 | 5 | 51 | 0 | 4 | 47 | 0.77 |

### TNF-α-308G/A(rs1800629)

| Number | Include studies | Race | Genotyping methods | Quality evaluation | Case/Control | Case | | | Control | | | $P_{HWE}$ |
|---|---|---|---|---|---|---|---|---|---|---|---|---|
| | | | | | | GG | GA | AA | GG | GA | AA | |
| 1 | Andra´ s Treszl 2003 | Europe | PCR-RFLP | 6 | 68/35 | 53 | 15 | 0 | 29 | 6 | 0 | 0.57 |
| 2 | Härtel C.[48] 2011 | Europe | ELISA | 6 | 354/1590 | 270 | 80 | 4 | 1127 | 427 | 36 | 0.55 |
| 3 | Härtel C.[48] 2011 | Europe | ELISA | 6 | 149/777 | 107 | 36 | 6 | 562 | 200 | 15 | 0.56 |
| 4 | A.C. Schuel ler 2006 | Europe | PCR | 6 | 169/233 | 199 | 85 | 34 | 68 | 29 | 5 | 0.42 |
| 5 | Curtis L. Hedberg, MD 2004 | Europe | PCR | 5 | 85/88 | 61 | 24 | | 59 | 29 | | not quite clear |
| 6 | GAMAL ALLAM 2015 | Asia | ELISA | 6 | 137/68 | 56 | 49 | 32 | 26 | 31 | 11 | 0.58 |
| 7 | Tatjana Varljen 2019 | Europe | PCR | 6 | 282/189 | 209 | 68 | 5 | 159 | 30 | 0 | 0.56 |
| 8 | Afdal Afdal 2022 | Asia | PCR | 5 | 30/30 | 28 | 2 | 0 | 20 | 10 | 0 | 0.27 |
| 9 | Suwannee Phumeetham 2012 | Asia | PCR | 6 | 66/101 | 58 | 8 | 0 | 86 | 13 | 2 | 0.09 |

**Table 2. Meta analysis related results.**

| | | IL-1βrs1143643 | | | | | | |
|---|---|---|---|---|---|---|---|---|

| Model | Number of studies | Heterogeneity test | | Effect model | Meta-analysis | | Begg's test (P) | Egger's test (P) |
|---|---|---|---|---|---|---|---|---|
| | | I²(%) | P | | OR(95%CI) | P | | |
| Allelic model (T vs C) | 4 | 92.9 | 0.000 | random | 0.506 (0.203–1.262) | 0.144 | 0.734 | 0.205 |
| | 2 | 0.0 | 0.450 | fixed | 0.224 (0.168–0.299) | 0.000 | 1.000 | - |
| Dominant gene model (TT+CT vs CC) | 4 | 92.4 | 0.000 | random | 0.771 (0.167–3.563) | 0.734 | 0.627 | 0.627 |
| | 2 | 56.6 | 0.129 | random | 2.785 (0.963–8.052) | 0.059 | 1.000 | - |
| Recessive gene model (TT vs CT+CC) | 4 | 89.9 | 0.000 | random | 0.791 (0.202–3.106) | 0.737 | 1.000 | 0.534 |
| | 3 | 19.3 | 0.290 | fixed | 1.567 (0.934–2.631) | 0.089 | 1.000 | 0.600 |
| Heterozygous gene model (TC vs CC) | 4 | 87.4 | 0.000 | random | 1.522 (0.393–5.904) | 0.543 | 0.734 | 0.964 |
| | 2 | 16.8 | 0.273 | fixed | 4.251 (2.226–8.119) | 0.000 | 1.000 | - |
| Homozygous gene model (TT vs CC) | 4 | 92.3 | 0.000 | random | 0.707 (0.092–5.452) | 0.740 | 0.734 | 0.606 |
| | 3 | 0.0 | 0.681 | fixed | 2.020 (1.122–3.639) | 0.019 | 1.000 | 0.831 |
| Additive gene model (TT+CC vs TC) | 4 | 85.0 | 0.000 | random | 0.904 (0.377–2.171) | 0.822 | 1.000 | 0.773 |

| | | IL6-174G/C | | | | | | |
|---|---|---|---|---|---|---|---|---|

| Model | Number of studies | Heterogeneity test | | Effect model | Meta-analysis | | Begg's test (P) | Egger's test (P) |
|---|---|---|---|---|---|---|---|---|
| | | I²(%) | P | | OR(95%CI) | P | | |
| Allelic model (C vs G) | 8 | 39.0 | 0.119 | fixed | 1.140 (0.974–1.335) | 0.101 | 0.536 | 0.349 |
| Dominant gene model (CC+GC vs GG) | 9 | 64.9 | 0.004 | random | 1.012 (0.689–1.488) | 0.950 | 0.348 | 0.701 |
| Recessive gene model (CC vs GC+GG) | 8 | 0.0 | 0.434 | fixed | 1.591 (1.154–2.194) | 0.005 | 0.386 | 0.273 |
| Heterozygous gene model (CG vs GG) | 8 | 72.1 | 0.001 | random | 0.968 (0.601–1.559) | 0.894 | 0.536 | 0.715 |
| | 6 | 47.4 | 0.091 | random | 1.021 (0.692–1.508) | 0.916 | 0.707 | 0.658 |
| Homozygous gene model (CC vs GG) | 8 | 0.0 | 0.686 | fixed | 1.471 (1.030–2.101) | 0.034 | 0.266 | 0.235 |
| Additive gene model (CC+GG vs CG) | 8 | 75.4 | 0.000 | random | 1.140 (0.714–1.821) | 0.583 | 0.536 | 0.790 |
| | 6 | 26.5 | 0.236 | fixed | 1.059 (0.835–1.343) | 0.635 | 0.707 | 0.809 |

| | | IL-8rs4073 | | | | | | |
|---|---|---|---|---|---|---|---|---|

| Model | Number of studies | Heterogeneity test | | Effect model | Meta-analysis | | Begg's test (P) | Egger's test (P) |
|---|---|---|---|---|---|---|---|---|
| | | I²(%) | P | | OR(95%CI) | P | | |
| Allelic model (T vs A) | 2 | 0.0 | 0.556 | fixed | 0.756 (0.549–1.041) | 0.087 | 1.000 | - |
| Dominant gene model (TT+AT vs AA) | 2 | 0.0 | 0.740 | fixed | 1.112(0.633–1.951) | 0.713 | 1.000 | - |
| Recessive gene model (TT vs AT+AA) | 2 | 0.0 | 0.571 | fixed | 0.467(0.280–0.777) | 0.003 | 1.000 | - |

(*Continued*)

**Table 2.** (Continued)

| Model | Number of studies | I²(%) | P | Effect model | OR(95%CI) | P | Begg's test (P) | Egger's test (P) |
|---|---|---|---|---|---|---|---|---|
| Heterozygous gene model (TA vs AA) | 2 | 0.0 | 0.938 | fixed | 1.548(0.855–2.806) | 0.149 | 1.000 | - |
| Homozygous gene model (TT vs AA) | 2 | 0.0 | 0.604 | fixed | 0.639(0.328–1.243) | 0.187 | 1.000 | - |
| Additive gene model (TT+AA vs TA) | 2 | 0.0 | 0.749 | fixed | 0.497(0.315–0.785) | 0.003 | 1.000 | - |

**IL-10−1082 G/A**

| Model | Number of studies | Heterogeneity test | | Effect model | Meta-analysis | | Begg's test (P) | Egger's test (P) |
|---|---|---|---|---|---|---|---|---|
| | | I²(%) | P | | OR(95%CI) | P | | |
| Allelic model (A vs G) | 6 | 75.6 | 0.001 | random | 1.090(0.691–1.719) | 0.712 | 0.260 | 0.335 |
| Dominant gene model (AA+GA vs GG) | 6 | 69.9 | 0.010 | random | 1.166(0.556–2.444) | 0.684 | 0.806 | 0.834 |
| | 5 | 0.0 | 0.759 | fixed | 1.731 (1.108–2.705) | 0.016 | 0.308 | 0.317 |
| Recessive gene model (AA vs GA+GG) | 6 | 55.9 | 0.045 | random | 1.248(0.728–2.138) | 0.420 | 0.133 | 0.147 |
| | 4 | 0.0 | 0.546 | fixed | 1.702 (1.218–2.377) | 0.002 | 1.000 | 0.776 |
| Heterozygous gene model (AA vs GG) | 6 | 40.9 | 0.149 | fixed | 1.110(0.739–1.666) | 0.616 | 0.806 | 0.676 |
| Homozygous gene model (AA vs GG) | 6 | 72.7 | 0.005 | random | 1.325(0.499–3.520) | 0.572 | 0.027 | 0.223 |
| | 5 | 0.0 | 0.716 | fixed | 2.282 (1.366–3.812) | 0.002 | 0.308 | 0.409 |
| Additive gene model (AA+GG vs AG) | 6 | 0.0 | 0.760 | fixed | 1.240(0.934–1.645) | 0.635 | 1.000 | 0.825 |

**TNF-α−308G/A(rs1800629)**

| Model | Number of studies | Heterogeneity test | | Effect model | Meta-analysis | | Begg's test (P) | Egger's test (P) |
|---|---|---|---|---|---|---|---|---|
| | | I²(%) | P | | OR(95%CI) | P | | |
| Allelic model (A vs G) | 8 | 66.2 | 0.004 | random | 1.057(0.787–1.419) | 0.712 | 1.000 | 0.963 |
| | 6 | 13.7 | 0.327 | fixed | 1.257 (1.044–1.513) | 0.016 | 1.000 | 0.719 |
| Dominant gene model (AA+GA vs GG) | 9 | 54.7 | 0.024 | random | 0.977(0.740–1.291) | 0.872 | 0.602 | 0.846 |
| | 8 | 43.2 | 0.090 | random | 1.023 (0.803–1.303) | 0.854 | 0.902 | 0.430 |
| Recessive gene model (AA vs GA+GG) | 8 | 52.7 | 0.061 | random | 1.357(0.659–2.796) | 0.408 | 1.000 | 0.913 |
| | 7 | 0.0 | 0.607 | fixed | 1.913 (1.179–3.104) | 0.009 | 0.806 | 0.954 |
| Heterozygous gene model (AA vs GG) | 8 | 50.8 | 0.047 | random | 0.944(0.706–1.261) | 0.695 | 0.902 | 0.810 |
| | 7 | 33.9 | 0.169 | fixed | 0.948(0.797–1.128) | 0.546 | 0.548 | 0.420 |
| Homozygous gene model (AA vs GG) | 8 | 41.2 | 0.131 | fixed | 1.341(0.881–2.043) | 0.171 | 1.000 | 0.994 |

(*Continued*)

**Table 2.** (Continued)

| Additive gene model (AA+GG vs AG) | 8 | 50.1 | 0.051 | random | 1.094(0.824–1.452) | 0.534 | 0.902 | 0.751 |
|---|---|---|---|---|---|---|---|---|
| | 7 | 33.4 | 0.173 | fixed | 1.079 (0.908–1.282) | 0.386 | 0.230 | 0.480 |

significant increase in the risk of neonatal sepsis with the AA+GA genotype (OR = 1.731, 95% CI: 1.108–2.705). Similarly, the OR value merged in the recessive gene model was statistically significant ($P = 0.002$), indicating a significant increase in the risk of sepsis in newborns with the AA genotype (OR = 1.702, 95% CI: 1.218–2.377). The OR value merged in the homozygous gene model was statistically significant ($P = 0.002$), indicating a significant increase in the risk of sepsis in newborns with the AA genotype (OR = 2.282, 95% CI: 1.366–3.812). No statistically significant difference was observed in the other gene models (allele model: $P = 0.260$, OR = 1.090, 95% CI: 0.691–1.719; heterozygous gene model: $P = 0.616$, OR = 1.110, 95% CI: 0.739–1.666; additive gene model: $P = 0.136$, OR = 1.240, 95% CI: 0.934–1.645).

Additionally, the results reveal that the OR value of TNF-α-308G/A (rs1800629) polymorphism merged the allele model was statistically significant ($P = 0.016$), indicating a significant increase in the risk of neonatal sepsis with the A allele (OR = 1.257, 95% CI: 1.044–1.513). Furthermore, the OR value merged in the recessive gene model was statistically significant ($P = 0.009$), indicating a significant increase in the risk of sepsis in newborns with the AA genotype (OR = 1.913, 95% CI: 1.179–3.10). There were no statistically significant differences between the other gene models (dominant gene model: $P = 0.854$, OR = 1.023, 95% CI: 0.803–1.303; heterozygous gene model: $P = 0.546$, OR = 0.948, 95% CI: 0.797–1.128; homozygous gene model: $P = 0.171$, OR = 1.341, 95% CI: 0.881–2.043; additive gene model: $P = 0.386$, OR = 1.079, 95% CI: 0.908–1.282).

### 3.6 Publication bias assessment (see Table 2)

The results of Begg's and Egger's tests show that the $P$-values of Begg's test for all gene models in IL-1βrs1143643 were greater than 0.05. However, due to the limited number of studies included in the allele display model, and heterozygous gene models, the Egger's results are not displayed. The $P$-values of the Egger's test for all other gene models were greater than 0.05. The $P$-values of Begg's and Egger's tests for all IL-6-174G/C gene models were greater than 0.05. Due to the limited number of studies included in all IL-8-rs4073 gene models, the Egger's results are not displayed, but the $P$-values of Begg's tests were all greater than 0.05. The $P$-value of Begg's test ($P = 0.027$) for the homozygous model of IL-10-1082 G/A is less than 0.05, but the $P$-value of Egger's test ($P = 0.223$) was greater than 0.05. The $P$-values of Begg's test and Egger's test for the other gene models were all greater than 0.05. The $P$-values of Begg's and Egger's tests for all gene models of TNF-α–308G/A(rs1800629)were greater than 0.05. Overall, we concluded that there was no significant publication bias in the included studies.

### 3.7 Subgroup analysis (see Table 3)

Subgroup analysis of studies with significant heterogeneity in the above results can be carried out from the aspects of age, sex, race, and whether they are in Hardy Weinberg equilibrium [37]. Since our study included full-term newborns and preterm newborns of different sexes, some studies did not provide detailed gestational age and sex information, and one study did not distinguish full-term and preterm newborns and their sexes; therefore, we did not use age

**Table 3. Subgroup analysis results.**

| IL-1βrs1143643 | | | | | | |
|---|---|---|---|---|---|---|
| **Model** | **Race(Effect *P*-value)** | | | **Compliance with HWE(Effect *P*-value)** | | |
| | Asia | Europe | All | No | Yes | All |
| Allelic model (T vs C) | 0.517 | 0.000 | 0.144 | 0.202 | 0.025 | 0.144 |
| Dominant gene model (TT+CT vs CC) | 0.608 | 0.011 | 0.739 | 0.012 | 0.335 | 0.739 |
| Recessive gene model (TT vs CT+CC) | 0.580 | 0.175 | 0.737 | 0.560 | 0.830 | 0.737 |
| Heterozygous gene model (TC vs CC) | 0.509 | 0.622 | 0.543 | 0.004 | 0.933 | 0.543 |
| Homozygous gene model (TT vs CC) | 0.613 | 0.049 | 0.740 | 0.166 | 0.533 | 0.740 |
| Additive gene model (TT+CC vs TC) | 0.985 | 0.112 | 0.822 | 0.006 | 0.532 | 0.822 |

| IL6-174G/C | | | | | | | |
|---|---|---|---|---|---|---|---|
| **Model** | **Race(Effect *P*-value)** | | | | **Compliance with HWE(Effect *P*-value)** | | |
| | Asia | Europe | Africa | All | No | Yes | All |
| Allelic model (C vs G) | 0.951 | 0.123 | 0.307 | 0.101 | 0.111 | 0.018 | 0.101 |
| Dominant gene model (CC+GC vs GG) | 0.009 | 0.429 | 0.359 | 0.950 | 0.852 | 0.835 | 0.950 |
| Recessive gene model (CC vs GC+GG) | 0.006 | 0.107 | 0.479 | 0.005 | 0.922 | 0.003 | 0.005 |
| Heterozygous gene model (CG vs GG) | 0.001 | 0.632 | 0.472 | 0.894 | 0.023 | 0.751 | 0.894 |
| Homozygous gene model (CC vs GG) | 0.310 | 0.097 | 0.343 | 0.034 | 0.388 | 0.009 | 0.034 |
| Additive gene model (CC+GG vs CG) | 0.000 | 0.871 | 0.733 | 0.583 | 0.039 | 0.859 | 0.583 |

| IL8-rs4073 | | | | | | |
|---|---|---|---|---|---|---|
| **Model** | **Race(Effect *P*-value)** | | | **Compliance with HWE(Effect *P*-value)** | | |
| | Asia | Europe | All | No | Yes | All |
| Allelic model (T vs A) | 0.497 | 0.094 | 0.087 | 0.094 | 0.497 | 0.087 |
| Dominant gene model (TT+AT vs AA) | 0.622 | 0.937 | 0.713 | 0.937 | 0.622 | 0.713 |
| Recessive gene model (TT vs AT+AA) | 0.132 | 0.011 | 0.003 | 0.011 | 0.132 | 0.003 |
| Heterozygous gene model (TA vs AA) | 0.329 | 0.285 | 0.149 | 0.285 | 0.329 | 0.149 |
| Homozygous gene model (TT vs AA) | 0.643 | 0.178 | 0.187 | 0.178 | 0.643 | 0.187 |
| Additive gene model (TT+AA vs TA) | 0.086 | 0.013 | 0.003 | 0.013 | 0.086 | 0.003 |

| IL-10–1082 G/A | | | | | | | |
|---|---|---|---|---|---|---|---|
| **Model** | **Race(Effect *P*-value)** | | | | **Compliance with HWE(Effect *P*-value)** | | |
| | Asia | Europe | Africa | All | No | Yes | All |

*(Continued)*

| Allelic model (A vs G) | 0.768 | 0.007 | 0.868 | 0.712 | 0.868 | 0.771 | 0.712 |
|---|---|---|---|---|---|---|---|
| Dominant gene model (AA+GA vs GG) | 0.785 | 0.035 | 0.768 | 0.684 | 0.768 | 0.747 | 0.684 |
| Recessive gene model (AA vs GA+GG) | 0.780 | 0.021 | 0.959 | 0.420 | 0.959 | 0.473 | 0.420 |
| Heterozygous gene model (AG vs GG) | 0.249 | 0.129 | 0.771 | 0.616 | 0.771 | 0.671 | 0.616 |
| Homozygous gene model (AA vs GG) | 0.785 | 0.006 | 0.804 | 0.572 | 0.804 | 0.645 | 0.572 |
| Additive gene model (AA+GG vs AG) | 0.058 | 0.558 | 0.871 | 0.136 | 0.871 | 0.095 | 0.136 |

| TNF-α−308G/A(rs1800629) | | | | | | |
|---|---|---|---|---|---|---|
| Model | Race(Effect *P*-value) | | | Compliance with HWE(Effect *P*-value) | | |
| | Asia | Europe | All | No | Yes | All |
| Allelic model (A vs G) | 0.344 | 0.351 | 0.712 | - | 0.712 | 0.712 |
| Dominant gene model (AA+GA vs GG) | 0.245 | 0.658 | 0.872 | - | 0.986 | 0.872 |
| Recessive gene model (AA vs GA+GG) | 0.527 | 0.474 | 0.408 | - | 0.408 | 0.408 |
| Heterozygous gene model (AG vs GG) | 0.199 | 0.776 | 0.695 | - | 0.695 | 0.695 |
| Homozygous gene model (AA vs GG) | 0.685 | 0.169 | 0.171 | - | 0.171 | 0.171 |
| Additive gene model (AA+GG vs AG) | 0.161 | 0.916 | 0.534 | - | 0.534 | 0.534 |

and sex for subgroup analysis. Given that the countries in the included studies were ethnically diverse, race may had an impact on the results; therefore, an ethnic subgroup analysis was performed for studies with high heterogeneity. We conducted HWE tests on the included studies, and a subgroup analysis was performed with a included studies with $P>0.05$ as the dividing line.

**From the perspective of race:** (1) In the allele model of IL-1βrs1143643 study, the *P*-values for Asian, European and global effects were 0.517, 0.000 and 0.144, respectively. The *P*-values for European were less than 0.05, but for Asian and global effects were greater than 0.05, indicating a need for sensitivity analysis. In the dominant gene model of the IL-1βrs1143643 study, the *P*-values for Asian, European and global effects were 0.608, 0.011 and 0.739, respectively. The *P*-values for European were less than 0.05, but for Asian and global effects were greater than 0.05, indicating a need for sensitivity analysis. In the recessive gene model of IL-1βrs1143643 study, the *P*-values of Asian, European and global effects were 0.580, 0.175 and 0.737, respectively, all greater than 0.05, indicating that IL-1βrs1143643 gene polymorphism was not associated with neonatal sepsis under the recessive gene model. In the heterozygous gene model of IL-1βrs1143643 study, the *P*-values for the Asian, European and global effects were 0.509, 0.622 and 0.543, respectively, all greater than 0.05, indicating that IL-1βrs1143643 gene polymorphism was not associated with neonatal sepsis under the heterozygous gene model. In the homozygous gene model of IL-1βrs1143643 study, the *P*-values for Asian, European and global effects were 0.613, 0.049 and 0.740, respectively. Although the *P*-values of European was less than 0.05, that of Asian and global effects were greater than 0.05, necessitating sensitivity analysis. Under the additive gene model of IL-1βrs1143643 in the study, the *P*-

values for Asian, European and global effects were 0.985, 0.112 and 0.822, respectively, all greater than 0.05, indicating that IL-1βrs1143643 gene polymorphism was not associated with neonatal sepsis under the additive gene model.

1. In the allele model of IL-6-174G/C study, the *P*-values for Asian, European, Africans and global effects were 0.951, 0.123, 0.307 and 0.101, respectively, which were all greater than 0.05, indicating that IL-6-174G/C gene polymorphism was not associated with neonatal sepsis under the allele model. In the dominant gene model of the IL-6-174G/C study, the *P*-values for Asian, European, African and global effects were 0.009, 0.429, 0.359 and 0.950, respectively. While the *P*-values of Asians effects were less than 0.05, and those of European, African and global effects were all greater than 0.05, requiring sensitivity analysis. Under the recessive gene model in IL-6-174G/C study, the *P*-values for Asian, European, African and global effects were 0.006, 0.107, 0.479 and 0.005, respectively. While the *P*-values for Asian and global effects were less than 0.05, those of European and African effects were greater than 0.05, requiring sensitivity analysis. In the heterozygous gene model of IL-6-174G/C study, the *P*-values for Asian, European, African and global effects were 0.001, 0.632, 0.472 and 0.894, respectively. Similar to the previous models, the *P*-values for Asian effects were less than 0.05, whereas those of European, African and global effects were greater than 0.05, requiring sensitivity analysis. In the homozygous gene model of IL-6-174G/C study, the *P*-values of Asian, European, African and global effects were 0.310, 0.097, 0.343 and 0.034, respectively. While the *P*-values of Asian, European, and African effects were greater than 0.05, the *P*-values of global effects were less than 0.05, requiring sensitivity analysis. In the additive gene model of the IL-6-174G/C study, the *P*-values for Asian, European, African and global effects were 0.000, 0.871, 0.733 and 0.583, respectively. The *P*-values of European, African, and global effects were greater than 0.05, but the *P*-values of Asian were less than 0.05, requiring sensitivity analysis.

2. In the allele model of IL-8-rs4073 study, the *P*-values for Asian, European and the global effects were 0.497, 0.094 and 0.087, respectively, which were all greater than 0.05, indicating that there was no correlation between IL-8-rs4073 gene polymorphism and neonatal sepsis under the allele model. Similarly, in the dominant gene model of the IL-8-rs4073 study, the *P*-values for Asian, European and the global effects were 0.622, 0.937 and 0.713, respectively, which were all greater than 0.05, indicating no association between the IL-8-rs4073 gene polymorphism and neonatal sepsis under the dominant gene model. In the recessive gene model of IL-8-rs4073 study, the *P*-values for Asian, European and global effects were 0.132, 0.011 and 0.003, respectively. Although, the *P*-values of Asian effects were greater than 0.05, those for European and global effects were less than 0.05, requiring sensitivity analysis. In the heterozygous gene model of IL-8-rs4073 study, the *P*-values for Asian, European and global effects were 0.329, 0.285 and 0.149, respectively, which were all greater than 0.05, indicating that IL-8-rs4073 gene polymorphism was not associated with neonatal sepsis under the heterozygous gene model. Likewise, in the homozygous gene model of IL-8-rs4073 study, the *P*-values for Asian, European and global effects were 0.643, 0.178 and 0.187, respectively, which were all greater than 0.05,suggesting that IL-8-rs4073 gene polymorphism was not associated with neonatal sepsis in the homozygous gene model. Under the additive gene model of the IL-8-rs4073 study, the *P*-values for Asian, and European and global effects were 0.086, 0.013 and 0.003, respectively. While the *P*-values of Asian effects were greater than 0.05, the values of the European and global effects were less than 0.05, requiring sensitivity analysis.

3. In the allele model of IL-10-1082 G/A study, the *P*-values of Asian, European, African and global effects were 0.768, 0.007, 0.868 and 0.712, respectively. While the *P*-values of European effects were less than 0.05, those of Asian, African and global effects were greater than 0.05, requiring sensitivity analysis. Similarly, under the dominant gene model of the IL-10-1082 G/A study, the *P*-values for Asian, European, African and global effects were 0.785, 0.035, 0.768 and 0.684, respectively. The *P*-values of European effects were less than 0.05, while those of Asian, African and global effects were greater than 0.05, requiring sensitivity analysis. Under the recessive gene model in IL-10-1082 G/A study, the *P*-values of Asian, European, African and global effects were 0.780, 0.021, 0.959 and 0.420, respectively. The *P*-values of Asian, African and global effects were greater than 0.05, but those of the European were less than 0.05, requiring sensitivity analysis. In the heterozygous gene model of the IL-10-1082 G/A study, the *P*-values for Asian, European, African and global effects were 0.249, 0.129, 0.771 and 0.616, respectively, which were all greater than 0.05, indicating no correlation between IL-10-1082 G/A gene polymorphism and neonatal sepsis under the heterozygous gene model. In the homozygous gene model of IL-10-1082 G/A study, the *P*-values for Asian, European, African and global effects were 0.785, 0.006, 0.804 and 0.572, respectively. The *P*-values for the European population were less than 0.05, but those of Asian, African and global effects were greater than 0.05, requiring sensitivity analysis. Under the additive gene model of IL-10-1082 G/A study, the *P*-values of Asian, European, Africans and global effects were 0.058, 0.558, 0.871 and 0.136, respectively, which were all greater than 0.05, indicating that the IL-10-1082 G/A gene polymorphism was not associated with neonatal sepsis in the additive gene model.

4. In the allele model of TNF-α-308G/A (rs1800629) study, the *P*-values for Asian, European and global effects were 0.344, 0.351 and 0.712, respectively, whinch were all greater than 0.05, indicating that there was no correlation between TNF-α-308G/A (rs1800629) gene polymorphism and neonatal sepsis in the allele model. Also, in the dominant gene model of TNF-α-308G/A (rs1800629) study, the *P*-values for Asian, European and global effects were 0.245, 0.658 and 0.872, respectively, which were all greater than 0.05, indicating that there was no correlation between TNF-α-308G/A (rs1800629) gene polymorphism and neonatal sepsis in the dominant gene model. Under the recessive gene model of TNF-α-308G/A (rs1800629) study, the *P*-values for Asian, European and global effects were 0.527, 0.474 and 0.408, respectively, which were all greater than 0.05, indicating that there was no correlation between TNF-α-308G/A (rs1800629) gene polymorphism and neonatal sepsis under the recessive gene model. In the heterozygous gene model of TNF-α-308G/A (rs1800629) study, the *P*-values for Asian, European and global effects were 0.199, 0.776 and 0.695, respectively, which were all greater than 0.05, indicating that there was no correlation between TNF-α-308G/A (rs1800629) gene polymorphism and neonatal sepsis under the heterozygous gene model. In the homozygous gene model of TNF-α-308G/A (rs1800629) study, the *P*-values for Asian, European and global effects were 0.685, 0.169 and 0.171, respectively, which were all greater than 0.05, indicating that there was no correlation between TNF-α-308G/A (rs1800629) gene polymorphism and neonatal sepsis under the homozygous gene model. Under the additive gene model of TNF-α-308G/A (rs1800629) study, the *P*-values for Asian, European and global effects were 0.161, 0.916 and 0.534, respectively, which were all greater than 0.05, indicating that there was no correlation between TNF-α-308G/A (rs1800629) gene polymorphism and neonatal sepsis under the additive gene model.

**From the perspective of HWE:** (1) In the allele model of IL-1βrs1143643 study, the *P*-values for non-conforming HWE, conforming HWE, and the overall effects were 0.202, 0.025

and 0.144, respectively. The *P*-values of non-conforming HWE (Mustarim Mustarim) and the overall effects were greater than 0.05, but conforming HWE effects were less than 0.05, requiring sensitivity analysis. In the dominant gene model of IL-1βrs1143643 study, the *P*-values of non-conforming HWE, conforming HWE and the overall effects were 0.012, 0.335 and 0.739, respectively. The *P*-values for non-conforming HWE effects were less than 0.05, while those of conforming HWE and overall effects were greater than 0.05, requiring sensitivity analysis. In the recessive gene model of IL-1βrs1143643 study, the *P*-values of non-conforming HWE, conforming HWE and the overall effects were 0.560, 0.830 and 0.737, respectively, which were all greater than 0.05, indicating that IL-1βrs1143643 gene polymorphism was not associated with neonatal sepsis under the recessive gene model. In the heterozygous gene model of IL-1βrs1143643 study, the *P* values of non-conforming HWE, conforming HWE and the overall effects were 0.004, 0.933 and 0.543, respectively. The *P* value of non-conforming HWE effects was less than 0.05, but conforming HWE and the overall effects were greater than 0.05, requiring sensitivity analysis. In the homozygous gene model of IL-1βrs1143643 study, the *P*-values for non-conforming HWE, conforming HWE and the overall effects were 0.166, 0.533 and 0.740, respectively, which were all greater than 0.05, indicating that there was no correlation between IL-1βrs1143643 gene polymorphism and neonatal sepsis under the homozygous gene model. In the additive gene model of study IL-1βrs1143643, the *P* values for non-conforming HWE, conforming HWE and the overall effects were 0.006, 0.532 and 0.822, respectively. The *P* value for non-conforming HWE effects was less than 0.05, but conforming HWE and the overall effects were greater than 0.05, requiring sensitivity analysis.

1. In the allele model of IL-6-174G/C study, the *P*-values of non-conforming HWE, conforming HWE and the overall effects were 0.111, 0.018 and 0.101, respectively. The *P*-values of non-conforming HWE (PETER AHRENS [37]) and the overall effects were greater than 0.05, while those of conforming HWE were less than 0.05, requiring sensitivity analysis. In the dominant gene model of IL-6-174G/C study, the *P*-values for non-conforming HWE, conforming HWE and the overall effects were 0.852, 0.835 and 0.950, respectively, which were all greater than 0.05, indicating that IL-6-174G/C gene polymorphism was not correlated with neonatal sepsis in the dominant gene model. In the recessive gene model of IL-6-174G/C study, the *P*-values for non-conforming HWE, conforming HWE and the overall effects were 0.922, 0.003 and 0.005, respectively. The *P*-values for non-conforming HWE effects were greater than 0.05, but those of conforming HWE and the overall effects were less than 0.05, requiring sensitivity analysis. In the heterozygous gene model of IL-6-174G/C study, the *P*-values for non-conforming HWE, conforming HWE and the overall effects were 0.023, 0.751 and 0.894, respectively. The *P*-values for non-conforming HWE effects were less than 0.05, but conforming HWE and the overall effects were greater than 0.05, requiring sensitivity analysis. In the homozygous gene model of IL-6-174G/C study, the *P*-values for non-conforming HWE, conforming HWE and the overall effects were 0.388, 0.009 and 0.034, respectively. The *P*-values of conforming HWE and the overall effects were less than 0.05, but those of non-conforming HWE effects were greater than 0.05, requiring sensitivity analysis. In the additive gene model of IL-6-174G/C study, the *P*-values of non-conforming HWE, conforming HWE and the overall effects were 0.039, 0.859 and 0.583, respectively. The *P*-values for the conforming HWE effects were less than 0.05, but non-conforming HWE and the overall effects were greater than 0.05, requiring sensitivity analysis.

2. In the allele model of IL-8-rs4073 study, the *P*-values for non-conforming HWE, conforming HWE and the overall effects were 0.094, 0.497 and 0.087, respectively, which were all greater than 0.05, indicating that IL-8-rs4073 gene polymorphism was not associated with

neonatal sepsis in the allele model. In the dominant gene model of IL-8-rs4073 study, the *P*-values of non-conforming HWE, conforming HWE and the overall effects were 0.937, 0.622 and 0.713, respectively, which were all greater than 0.05, indicating that IL-8-rs4073 gene polymorphism was not correlated with neonatal sepsis in the dominant gene model. In the recessive gene model of IL-8-rs4073 study, the *P*-values for non-conforming HWE, conforming HWE and the overall effects were 0.011, 0.132 and 0.003, respectively. The *P*-values for HWE effect were greater than 0.05, while those of the non-conforming HWE and the overall effects were less than 0.05,requiring sensitivity analysis. In the heterozygous gene model of IL-8-rs4073 study, the *P*-values of non-conforming HWE, conforming HWE and the overall effects were 0.285, 0.329 and 0.149, respectively, which were all greater than 0.05, indicating that IL-8-rs4073 gene polymorphism was not associated with neonatal sepsis in the heterozygous gene model. In the homozygous gene model of IL-8-rs4073 study, the *P*-values of non-conforming HWE, conforming HWE and the overall effects were 0.178, 0.643 and 0.187, respectively, which were all greater than 0.05, indicating that IL-8-rs4073 gene polymorphism was not associated with neonatal sepsis in the homozygous gene model. Under the additive gene model of IL-8-rs4073 study, the *P*-values for non-conforming HWE, conforming HWE and the overall effects were 0.013, 0.086 and 0.003, respectively. The *P*-values of conforming HWE effects were greater than 0.05, while those of non-conforming HWE and the overall effects were less than 0.05, requiring sensitivity analysis.

3. In the allele model of IL-10-1082G/A study, the *P*-values of non-conforming HWE, conforming HWE and the overall effects were 0.868, 0.771 and 0.712, which were all greater than 0.05, indicating that IL-10-1082 G/A gene polymorphism was not associated with neonatal sepsis in the allele model. In the dominant gene model of IL-10-1082 G/A study, the *P*-values of non-conforming HWE, conforming HWE and the overall effects were 0.768, 0.747 and 0.684, which were all greater than 0.05, indicating that IL-10-1082 G/A gene polymorphism was not associated with neonatal sepsis in the dominant gene model. In the recessive gene model of IL-10-1082 G/A study, the *P*-values for non-conforming HWE, conforming HWE and the overall effects were 0.959, 0.473 and 0.420, which were all greater than 0.05, indicating that IL-10-1082 G/A gene polymorphism was not associated with neonatal sepsis in the recessive gene model. In the heterozygous gene model of IL-10-1082 G/A study, the *P*-values for non-conforming HWE, conforming HWE and the overall effects were 0.771, 0.671 and 0.616, which were all greater than 0.05, indicating that IL-10-1082 G/A gene polymorphism was not associated with neonatal sepsis in the heterozygous gene model. In the homozygous gene model of IL-10-1082 G/A study, the *P*-values of non-conforming HWE, conforming HWE and the overall effects were 0.804, 0.645 and 0.572, which were all greater than 0.05, indicating that IL-10-1082 G/A gene polymorphism was not associated with neonatal sepsis under the homozygous gene model. In the additive gene model of IL-10-1082 G/A study, the *P*-values of non-conforming HWE, conforming HWE and the overall effects were 0.871, 0.095, and 0.136, which were all greater than 0.05, indicating that IL-10-1082 G/A gene polymorphism was not associated with neonatal sepsis under the additive gene model.

4. In the allele model of TNF-α-308G/A (rs1800629) study, the *P*-values of conforming HWE and the overall effects were 0.712 and 0.712, respectively, which were greater than 0.05, indicating that there was no correlation between TNF-α-308G/A (rs1800629) gene polymorphism and neonatal sepsis under the allele model. In the dominant gene model of TNF-α-308G/A (rs1800629) study, the *P*-values of conforming HWE and the overall effects were 0.986 and 0.872, respectively, which were greater than 0.05, indicating that there was

no correlation between TNF-α-308G/A (rs1800629) gene polymorphism and neonatal sepsis under the dominant gene model. In the recessive gene model in TNF-α-308G/A (rs1800629) study, the *P*-values of the conforming HWE and the overall effects were 0.408 and 0.408, respectively, which were greater than 0.05, indicating that there was no correlation between TNF-α-308G/A (rs1800629) gene polymorphism and neonatal sepsis under the recessive gene model. In the heterozygous gene model of TNF-α-308G/A (rs1800629) study, the *P*-values of conforming HWE and the overall effects were 0.695 and 0.695, respectively, which were greater than 0.05, indicating that there was no correlation between TNF-α-308G/A (rs1800629) gene polymorphism and neonatal sepsis under the heterozygous gene model. In the homozygous gene model of TNF-α-308G/A (rs1800629) study, the *P*-values of conforming HWE and the overall effects were 0.171 and 0.171, respectively, which were greater than 0.05, indicating that there was no correlation between TNF-α-308G/A (rs1800629) gene polymorphism and neonatal sepsis under the homozygous gene model. In the additive gene model of TNF-α-308G/A (rs1800629) study, the *P*-values of the conforming HWE and the overall effects were 0.534 and 0.534, which were greater than 0.05, indicating that there was no correlation between TNF-α-308G/A (rs1800629) gene polymorphism and neonatal sepsis under the additive gene model.

## 3.8 Sensitivity analysis

One study was removed at a time, and the pooled OR values for the remaining studies were calculated for sensitivity analysis to identify the sources of heterogeneity. According to the results, four studies related to IL-1βrs1143643 were deleted under the allele model, two studies (Mustarim Mustarim and Zhao Xiaofen) were deleted, and then meta-analysis was performed, and heterogeneity disappeared ($I^2 = 0.00\%$, $P = 0.450$). In the dominant gene model, two studies (GAMAL ALLAM and Zhao Xiaofen) were removed, and the heterogeneity disappeared ($I^2 = 0.00\%$, $P = 0.596$). In the recessive gene model, one study (GAMAL ALLAM) was removed, and the heterogeneity disappeared ($I^2 = 19.3\%$, $P = 0.290$). In the heterozygous gene model, two studies (Susanna Esposito and Zhao Xiaofen) were removed, and the heterogeneity disappeared ($I^2 = 16.8\%$, $P = 0.273$). In the homozygous gene model, heterogeneity disappeared after the removal of one study (GAMAL ALLAM) ($I^2 = 0.00\%$, $P = 0.681$). Under the additive gene model, the heterogeneity was significant, and there was no significant change after the removal of the study ($I^2 = 85.0\%$, $P = 0.000$).

In the nine studies involving IL-6-174G/C, the heterogeneity was significant in the dominant gene model, and there was no significant change after removing the study ($I^2 = 64.9\%$, $P = 0.004$). In the heterozygous and additive gene models, two studies (RJohnBaier and GAMAL ALLAM) were removed, and the heterogeneity disappeared ($I^2 = 47.4\%$, $P = 0.091$; additive gene model: $I^2 = 26.5\%$, $P = 0.236$).

In the six studies involving IL-10-1082 G/A, there was significant heterogeneity in the allele model, and no significant change was observed after the removal of this study ($I^2 = 75.6\%$, $P = 0.001$). In the dominant and homozygous gene models, one study (Bao Jin) was removed, and the heterogeneity disappeared (dominant gene model: $I^2 = 0.00\%$, $P = 0.795$; homozygous gene model: $I^2 = 0.00\%$, $P = 0.716$). In the recessive gene model, two studies (Bao Jin and Zhao Xiaofen) were removed and the heterogeneity disappeared ($I^2 = 0.00\%$, $P = 0.546$).

Among the eight studies related to TNF-α-308G /A(rs1800629), in the dominant gene model, recessive gene model, heterozygous gene model and additive gene model, one study (Afdal Afdal) was removed, and the heterogeneity disappeared (dominant gene model: $I^2 = 43.2\%$, $P = 0.090$; recessive gene model: $I^2 = 0.00\%$, $P = 0.607$; heterozygous gene model: $I^2 =$

33.9%, $P$ = 0.169; additive gene model: $I^2$ = 33.4%, $P$ = 0.173). In the allele model, two studies (Härtel C. MD [48] and Afdal Afdal) were removed, and the heterogeneity disappeared ($I^2$ = 13.7%, $P$ = 0.327).

The results showed that in the above mentioned studies on cytokine gene polymorphisms, studies that had a greater impact on heterogeneity were deleted, and the combined effect variables of the final result analysis were large. After deleting studies with large heterogeneity, the random removal of one study did not change the results of the entire meta-analysis, but only slightly affected the combined effect size. The sensitivity analysis showed that the results of the meta-analysis were reliable and stable.

## 3.9 Sequential analysis and multiple test correction

**Sequential analysis (TSA):** Sequential analysis of included studies using TSA0.9.5.10Beta software [54–63], setting the probability of Class I errors α = 5%, Class II error probability β = 20%, relative risk reduction rate RRR = 20%.

About the TSA results of IL-1β rs1143643 showed that in the allele model, the cumulative Z-value of meta-analysis crossed both the traditional and TSA thresholds, indicating that although the cumulative information did not reach the expected value, no more experiments were needed to obtain a positive conclusion in advance(see Fig 6). In the dominant gene model, the cumulative Z-value of the meta-analysis crossed the traditional threshold but did not cross the TSA threshold, and its cumulative information did not reach the expected information, indicating the possibility of false-positive results, which still requires further verification (see Fig 7). In the recessive gene model, the cumulative Z-value of the meta-analysis did not cross the traditional threshold or TSA threshold, and the cumulative information did not reach the expected value, indicating any statistical significance and still requiring a large number of experiments for verification (see Fig 8). Due to the limited number of studies included in the heterozygous and homozygous gene models, the software automatically ignored the TSA threshold; therefore, there was no TSA threshold in the figure. The cumulative Z-value of

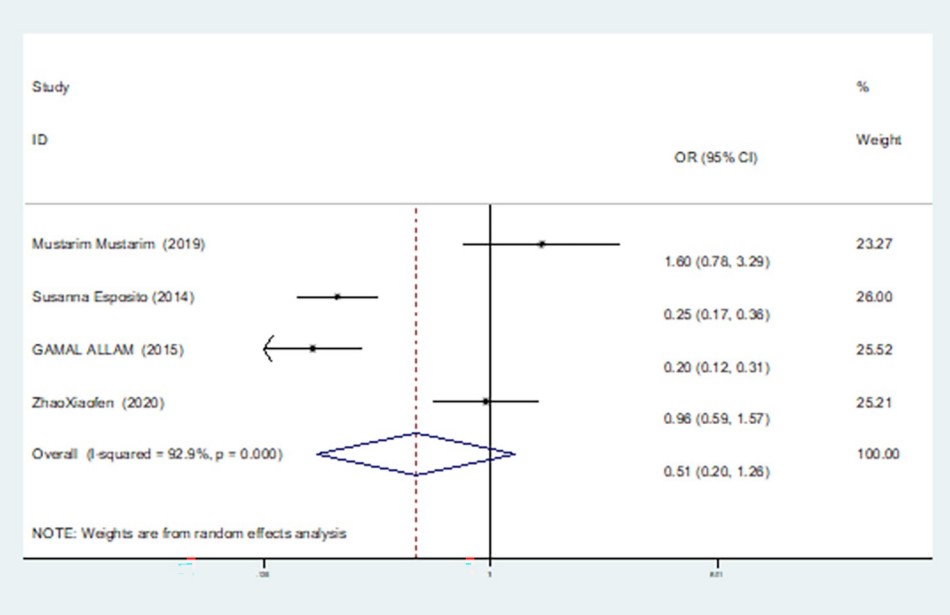

**Fig 6. TSA map of IL-1 allele.**

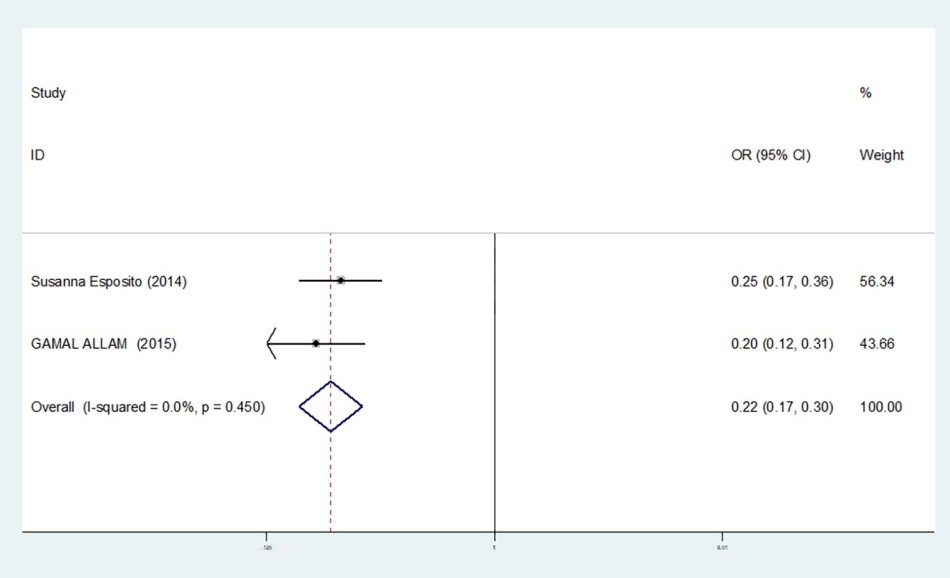

**Fig 7. TSA map of IL-1 dominant gene.**

the meta-analysis crossed the traditional threshold, and its cumulative information did not reach the expected information, indicating that false-positive results may have occurred, and more experiments are needed for verification (see Figs 9 and 10). In the additive gene model, because of the limited number of studies included, the software automatically ignored the TSA threshold; therefore, there was no TSA threshold in the figure. The cumulative Z-value of the meta-analysis did not cross the traditional threshold or TSA threshold, and the cumulative

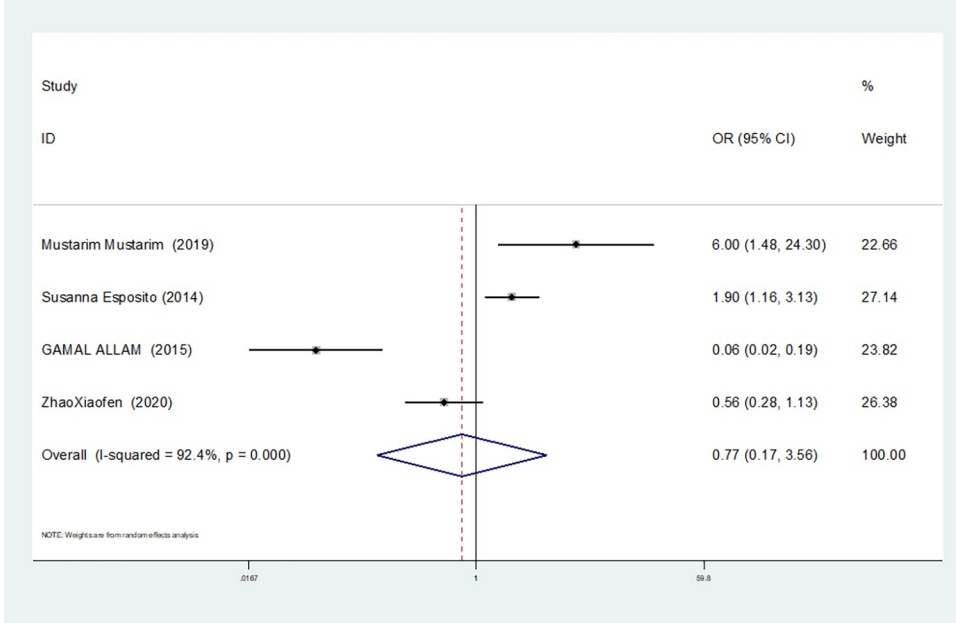

**Fig 8. TSA map of IL-1 recessive gene.**

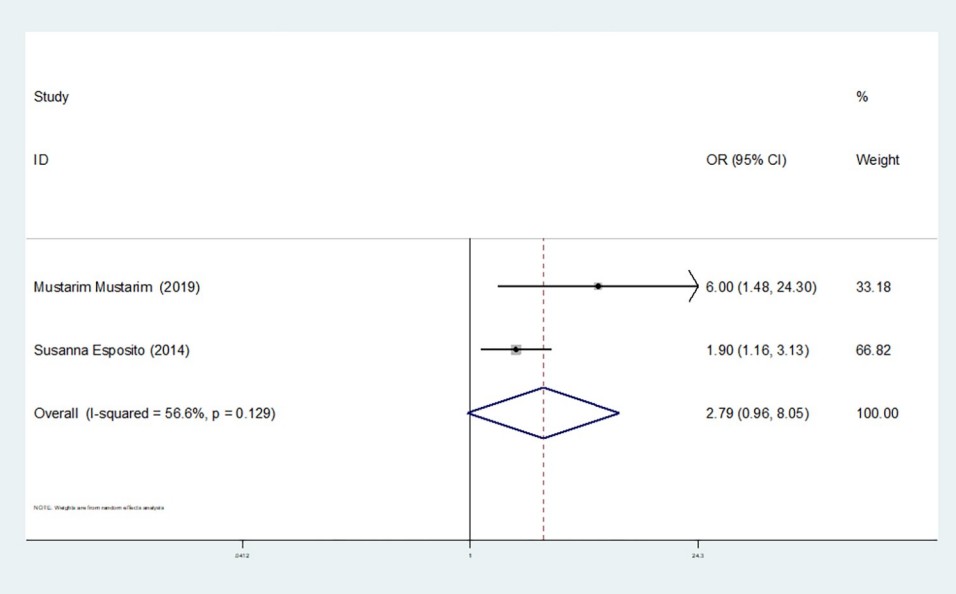

**Fig 9. TSA map of IL-1 heterozygous gene.**

information did not reach the expected value, indicating that it has no statistical significance and still needs to be verified through a large number of experiments (see Fig 11).

The TSA results of IL-6-174G/C showed that in the allele, dominant gene, heterozygous gene and additive gene models, the cumulative Z-value of the meta-analysis did not cross the traditional threshold or TSA threshold, and the cumulative information did not reach the expected value, indicating no statistical significance. Further experiments are required to verify

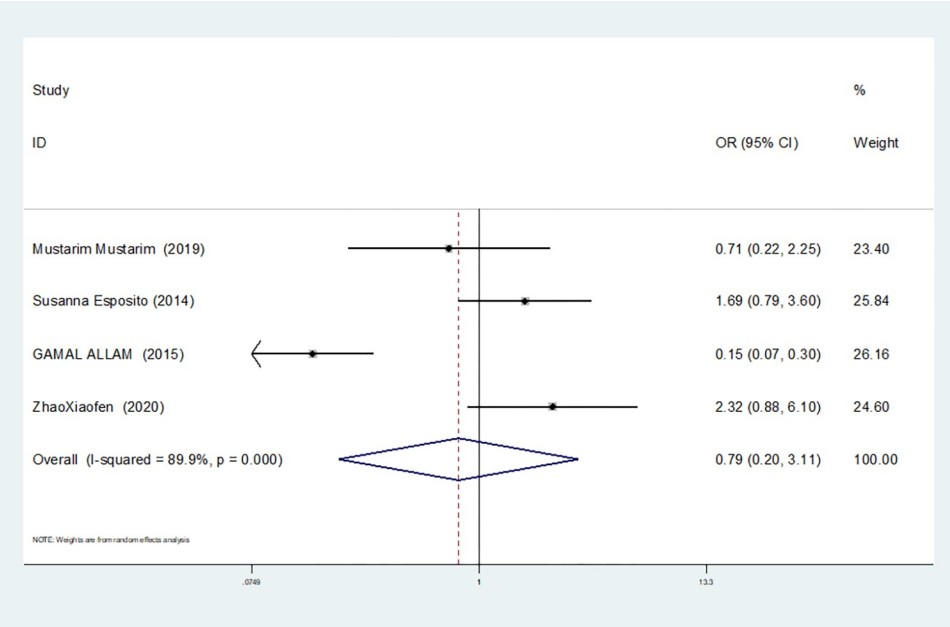

**Fig 10. TSA map of IL-1 homozygous gene.**

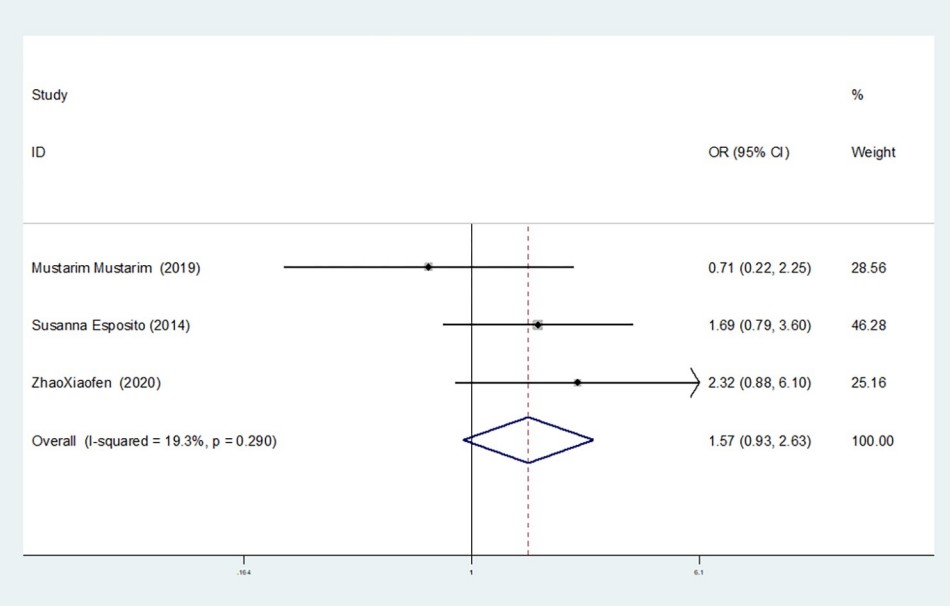

**Fig 11. TSA map of IL-1 additive gene.**

this (see Figs 12–15). In the recessive and homozygous gene models, the cumulative Z-value of the meta-analysis crossed the traditional threshold but not the TSA threshold, and its cumulative information did not reach the expected information level, indicating that false-positive results may have occurred in further experiments are needed to verify (see Figs 16 and 17).

The TSA results of IL-8-rs4073 showed that in both the allele model and the dominant gene model, the cumulative Z-value of the meta-analysis did not cross the traditional threshold or

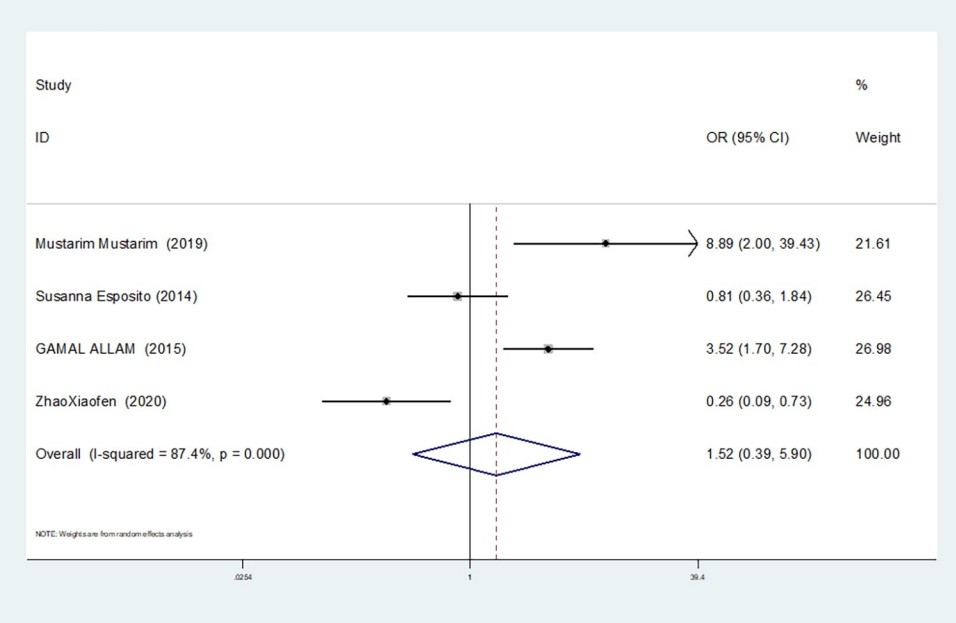

**Fig 12. TSA map of IL-6 allele.**

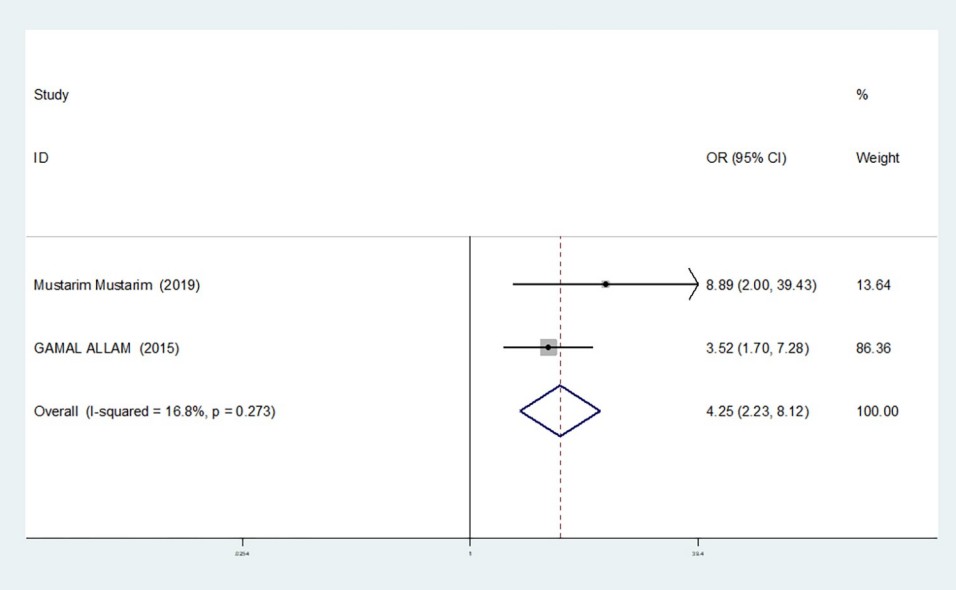

**Fig 13. TSA map of IL-6 dominant gene.**

TSA threshold, and the cumulative information did not reach the expected information level, indicating no statistical significance and still requiring a large number of experiments for verification (see Figs 18 and 19). In both the recessive gene model and the additive gene model, the cumulative Z-value of the meta-analysis crossed the traditional threshold but did not cross the TSA threshold, and its cumulative information did not reach the expected information level, indicating that false-positive results may have occurred. Further experiments are needed to verify these results (see Figs 20 and 21). In both the heterozygous and homozygous gene

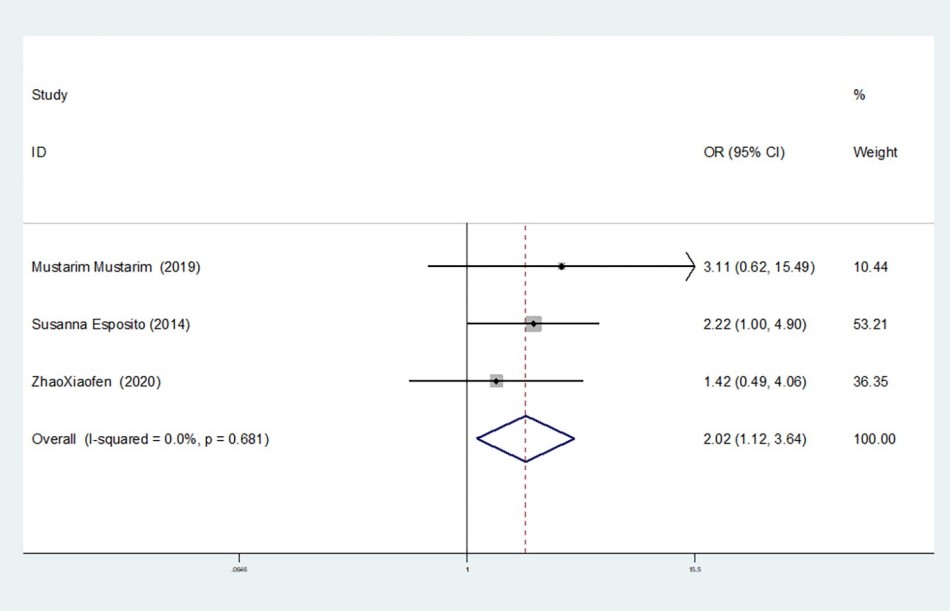

**Fig 14. TSA map of IL-6 heterozygous gene.**

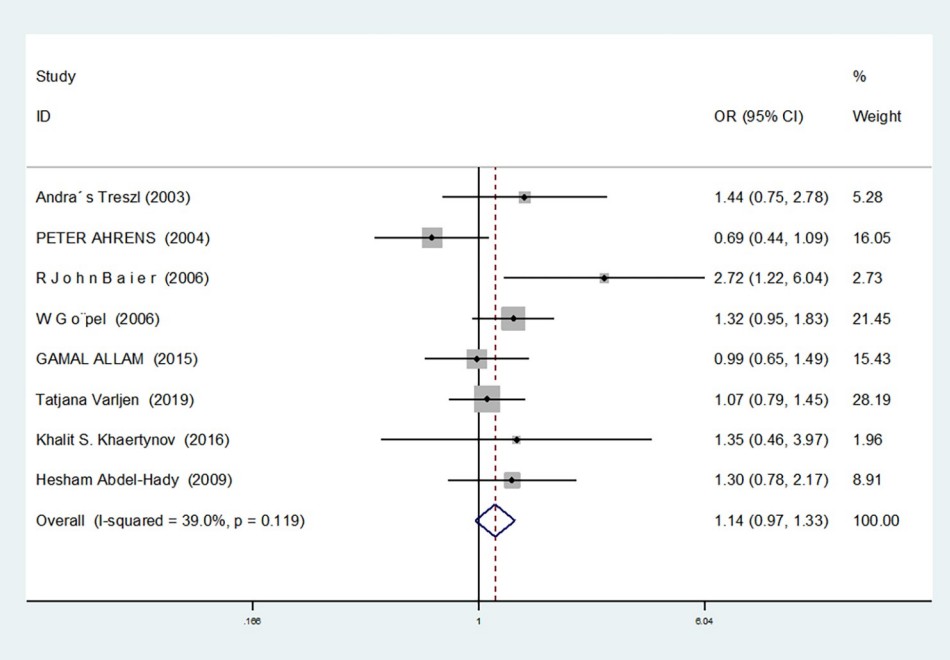

**Fig 15. TSA map of IL-6 additive gene.**

models, because of the limited number of studies included, the software automatically ignored the TSA threshold. Therefore, there was no TSA threshold in the graph. The cumulative Z-value of the meta-analysis did not cross the traditional threshold or TSA threshold, and the cumulative information did not reach the expected information, indicating any statistical significance. Further experiments are required to verify this (see Figs 22 and 23).

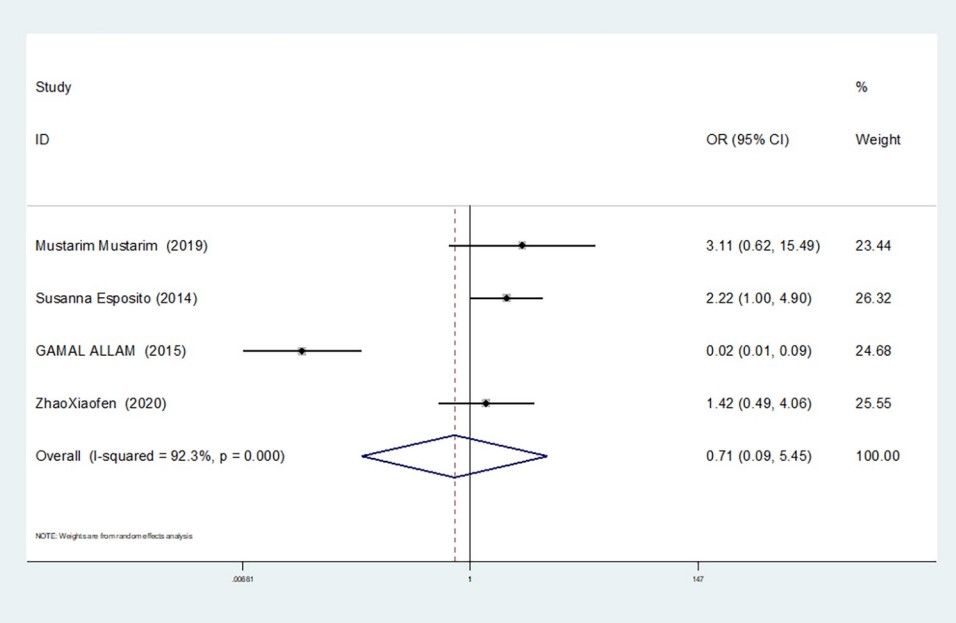

**Fig 16. TSA map of IL-6 recessive gene.**

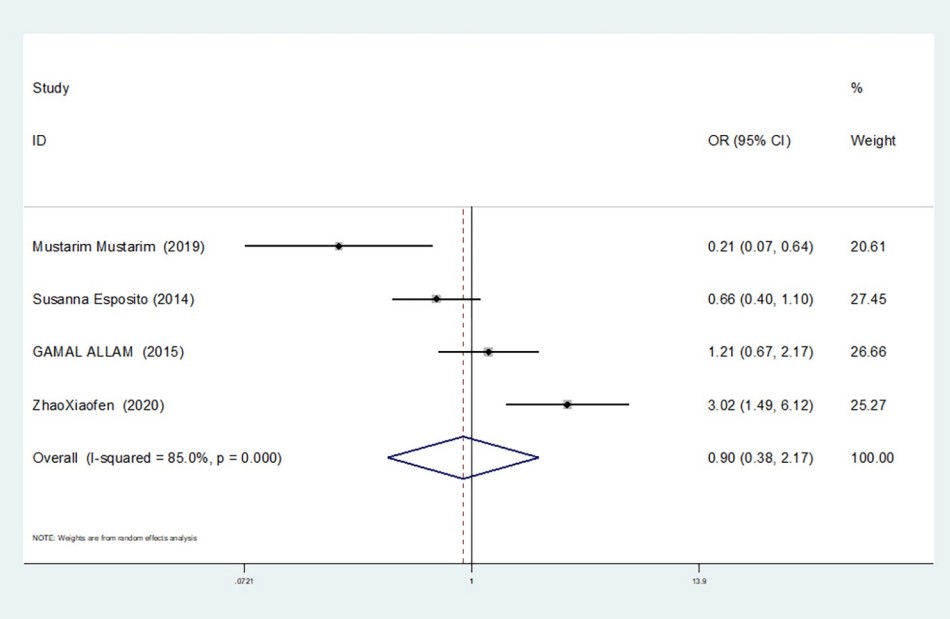

**Fig 17. TSA map of IL-6 homozygous gene.**

The TSA results for IL-10-1082G/A showed that in the allele, dominant gene and homozygous gene models, because of the limited number of studies included, the software automatically ignored the TSA threshold. Therefore, there was no TSA threshold in the figure. The cumulative Z-value of the meta-analysis crossed the traditional threshold, and its cumulative

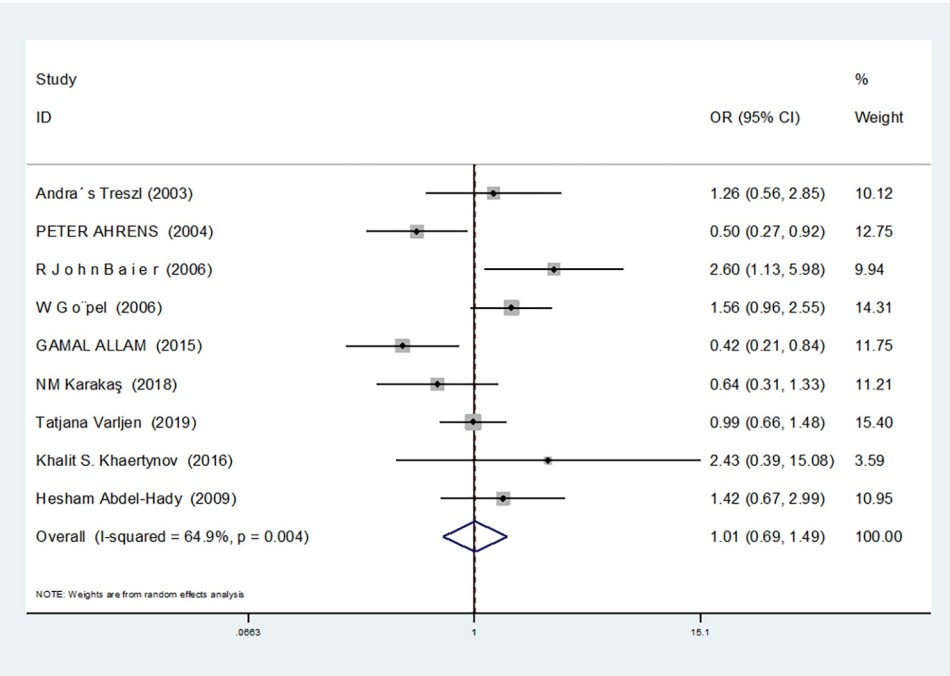

**Fig 18. TSA map of IL-8 allele.**

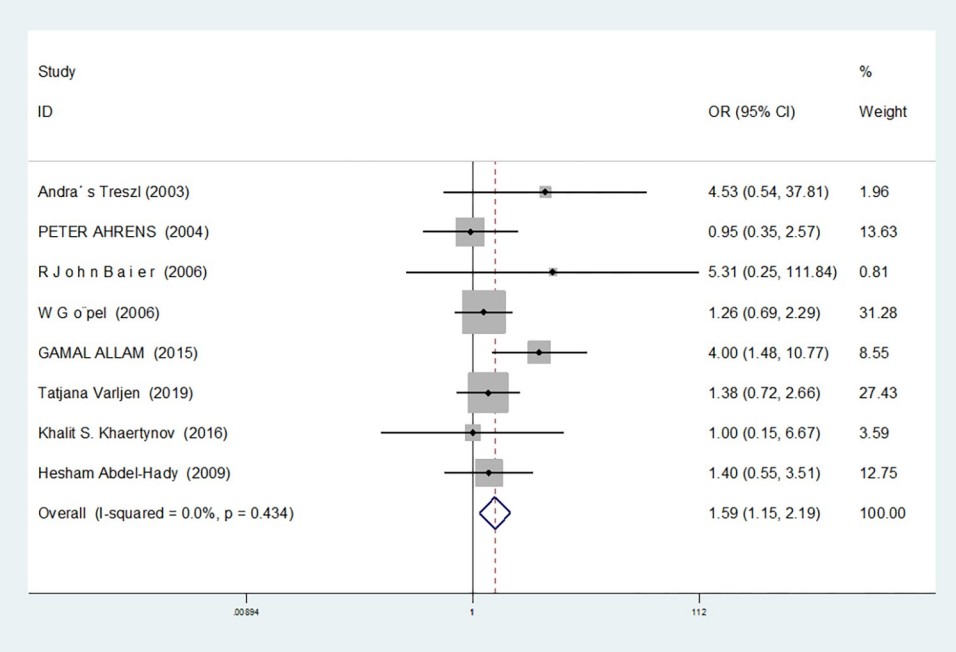

**Fig 19. TSA map of IL-8 dominant gene.**

information did not reach the expected value, indicating that false-positive results may have occurred, and more experiments are needed for verification (see Figs 24–26). In the recessive gene model, the cumulative Z-value of the meta-analysis crossed the traditional threshold but did not cross the TSA threshold, and its cumulative information did not reach the expected

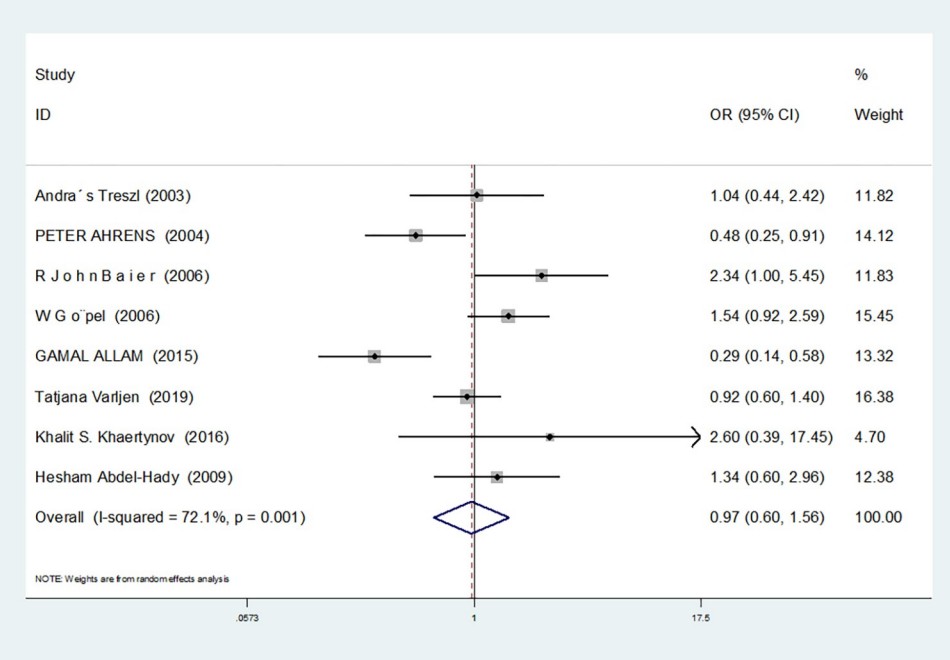

**Fig 20. TSA map of IL-8 recessive gene.**

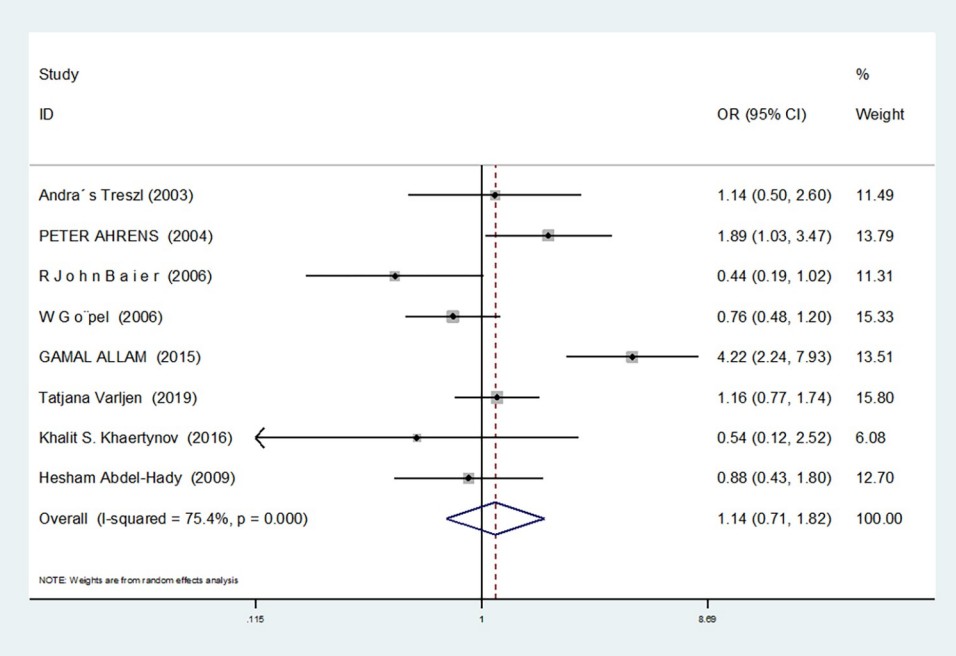

**Fig 21. TSA map of IL-8 additive gene.**

information, indicating that false-positive results may have occurred; further experiments are needed to verify this (see Fig 27). In the heterozygous and additive gene models, the cumulative Z-value of the meta-analysis did not cross the traditional threshold or TSA threshold, and the cumulative information did not reach the expected information, indicating any statistical

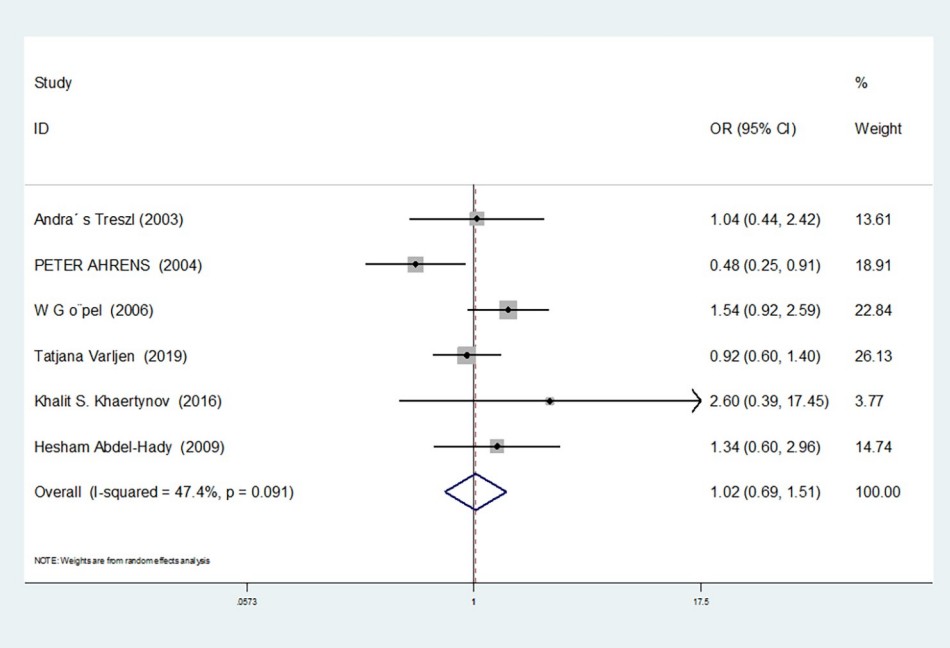

**Fig 22. TSA map of IL-8 heterozygous gene.**

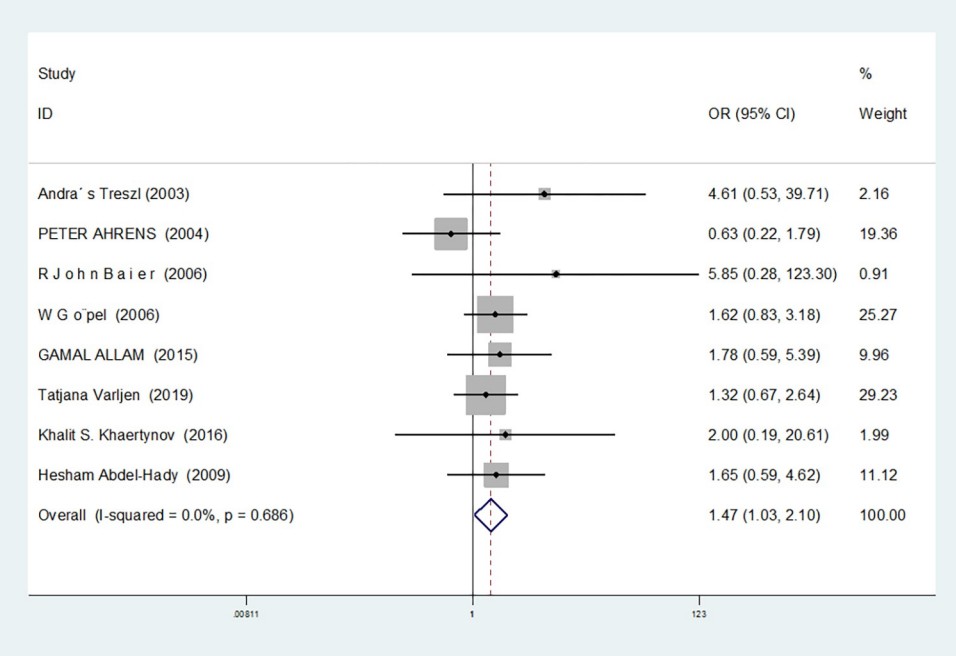

**Fig 23. TSA map of IL-8 homozygous gene.**

significance and still requiring a large number of experiments for verification (see Figs 28 and 29).

The TSA results of TNF-α-308G/A (rs1800629) showed that in the allele and recessive gene models, the cumulative Z-value of meta-analysis crossed the traditional threshold but did not

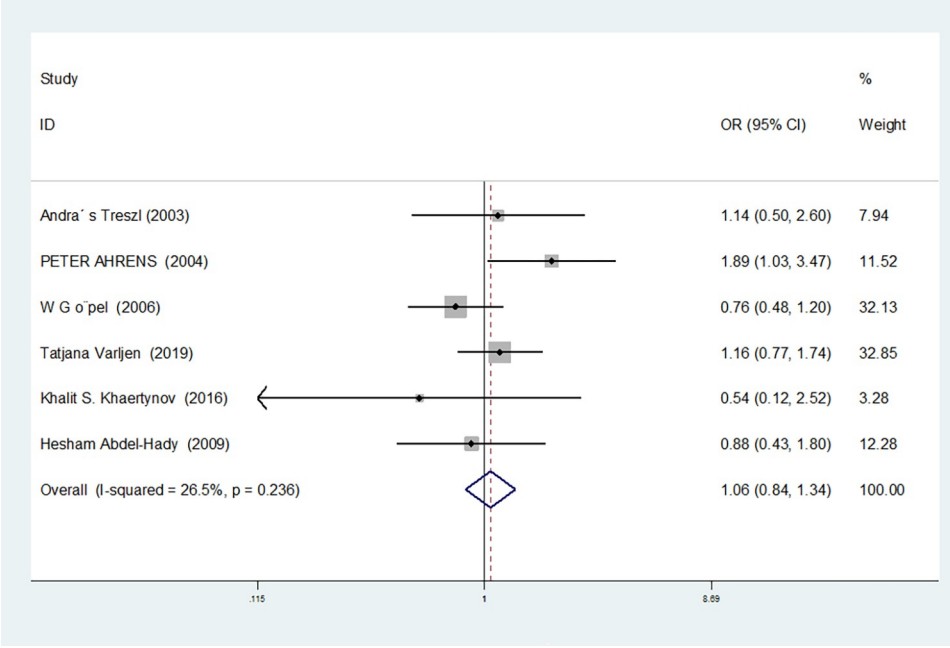

**Fig 24. TSA map of IL-10 allele.**

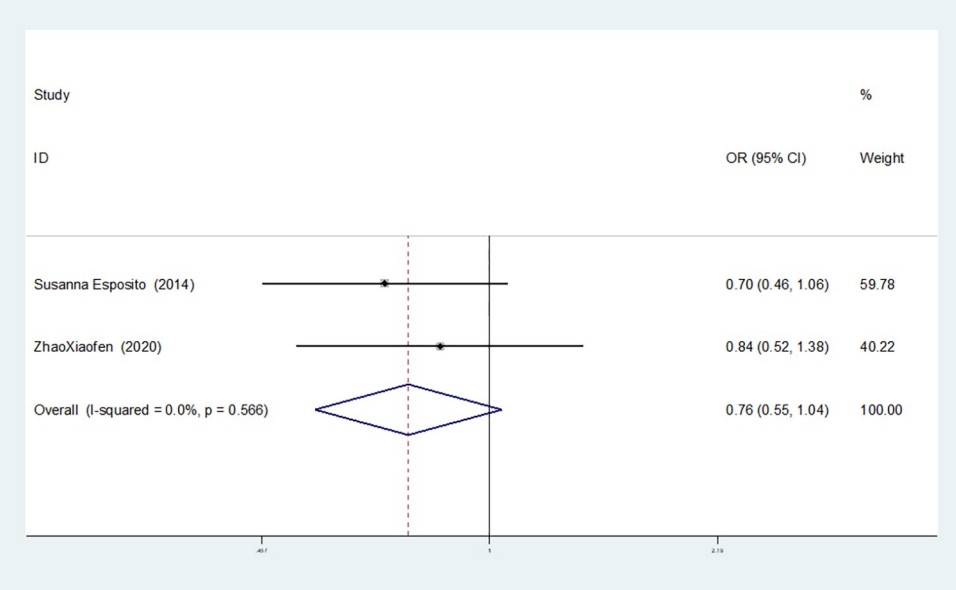

**Fig 25. TSA map of IL-10 dominant gene.**

cross the TSA threshold, and its cumulative information did not reach the expected information, indicating the possibility of false-positive results, which still needs more experiments to verify (see Figs 30 and 31). In the dominant gene model, heterozygous gene model, homozygous gene model and additive gene model, the cumulative Z-value of the meta-analysis did not cross the traditional threshold or TSA threshold, and the cumulative information did not reach the expected information level, indicating that there was no statistical significance. A large number of experiments are needed to verify this (see Figs 32–35).

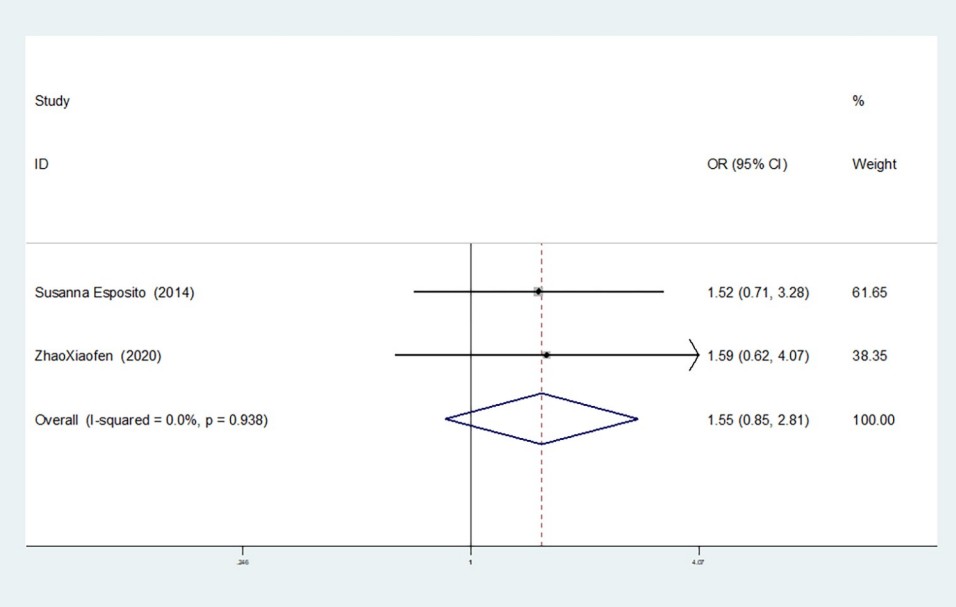

**Fig 26. TSA map of IL-10 homozygous gene.**

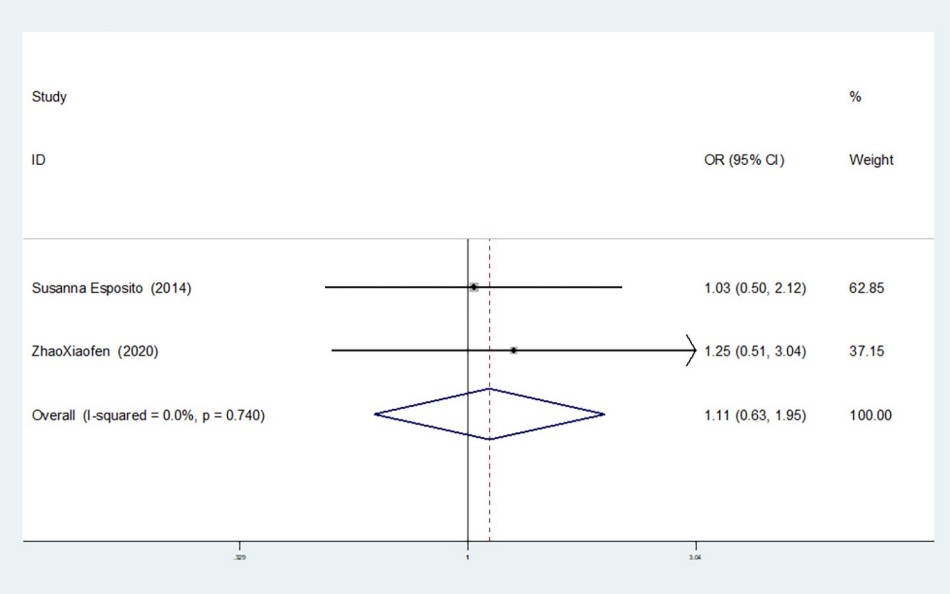

**Fig 27. TSA map of IL-10 recessive gene.**

**Multiple test correction:** Our study calculated the results of six genetic models and conducted multiple hypothesis tests, but our testing level α = 0.05 did not change, which significantly increased the probability of false-positive errors [64, 65]. Therefore, we used the Bonferroni correction and false-positive reporting rate (FPRP) methods to evaluate whether the research results were false-positives.

1. Bonferroni correction (results shown in Table 4): This meta-analysis tested six gene models, so our adjusted testing level α' = 0.05/6 ≈ 0.0083 [66], and then comparing the *P*-values of

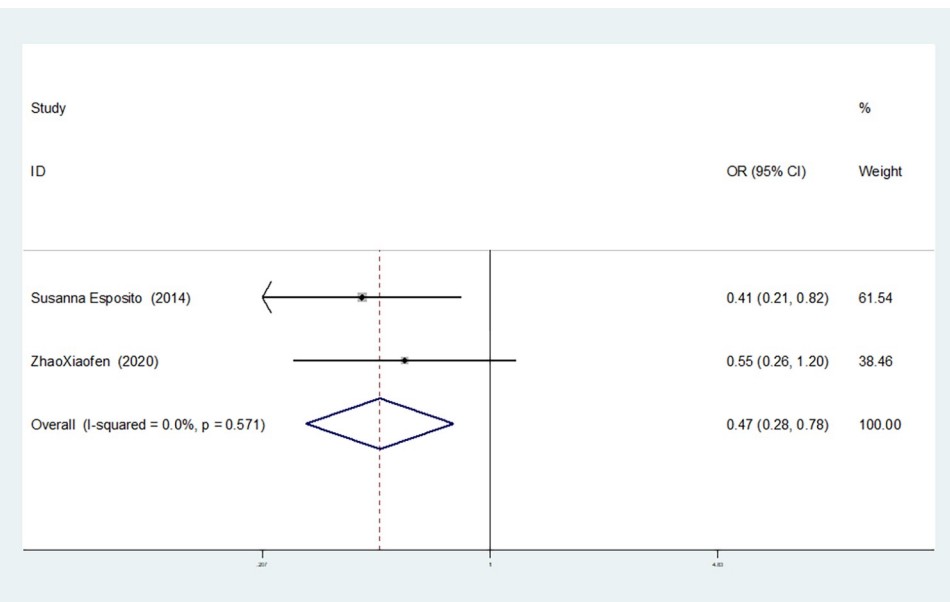

**Fig 28. TSA map of IL-10 heterozygous gene.**

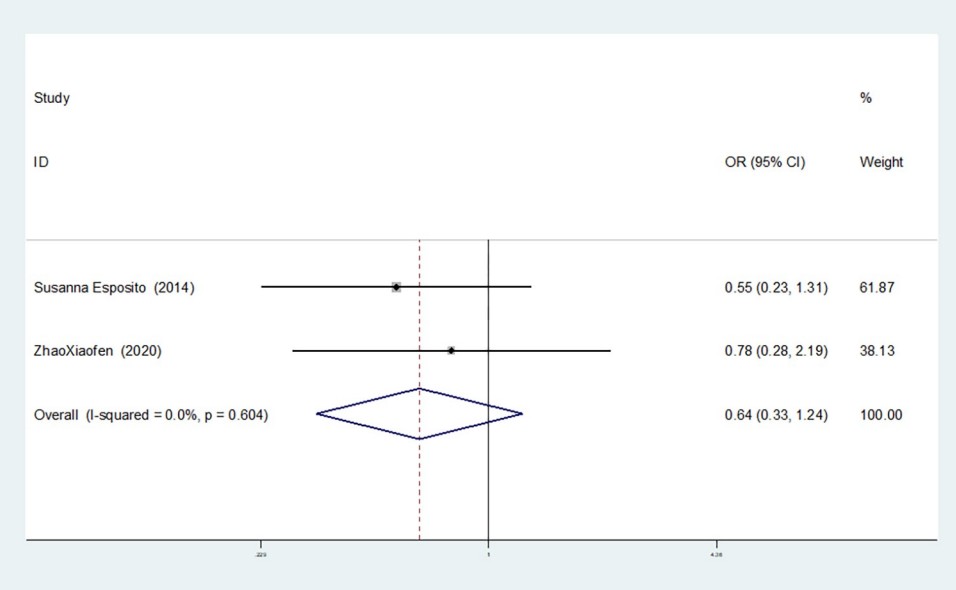

**Fig 29. TSA map of IL-10 additive gene.**

each gene model for each cytokine with 0.0083, we believe that $P<0.0083$ has statistical significance. The Bonferroni correction results of IL-1βrs1143643 showed that $P$ was less than 0.0083 in both the allele model ($P = 0.000$) and heterozygous gene model ($P = 0.000$), indicating statistical significance. In the homozygous gene model ($P = 0.019$), $P$ was less than 0.05 and greater than 0.0083, indicating the possibility of false-positive results. However, further studies are required to test this hypothesis. The Bonferroni correction results for IL-6-174G/C showed that $P$ was less than 0.0083 in the recessive gene model ($P = 0.005$),

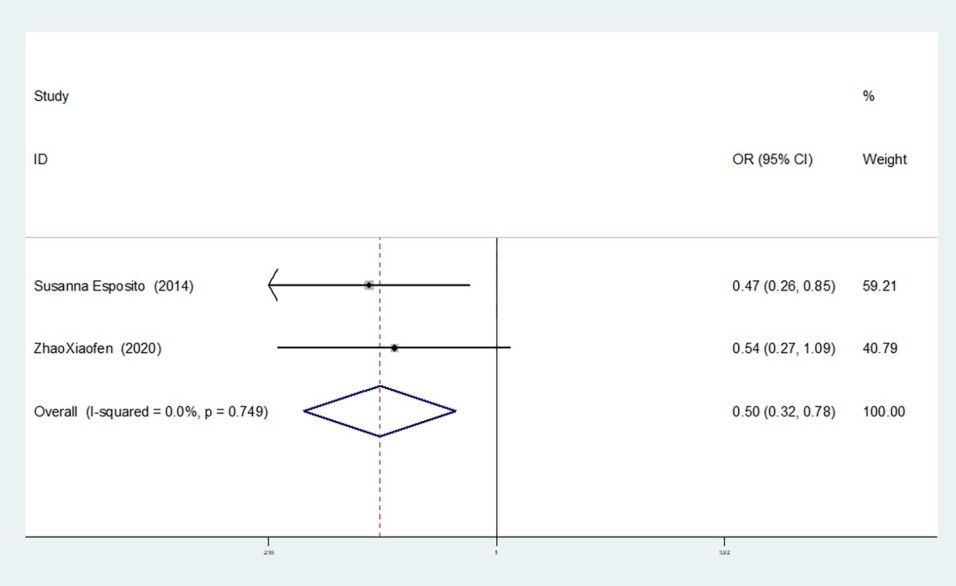

**Fig 30. TSA map of TNF-α allele.**

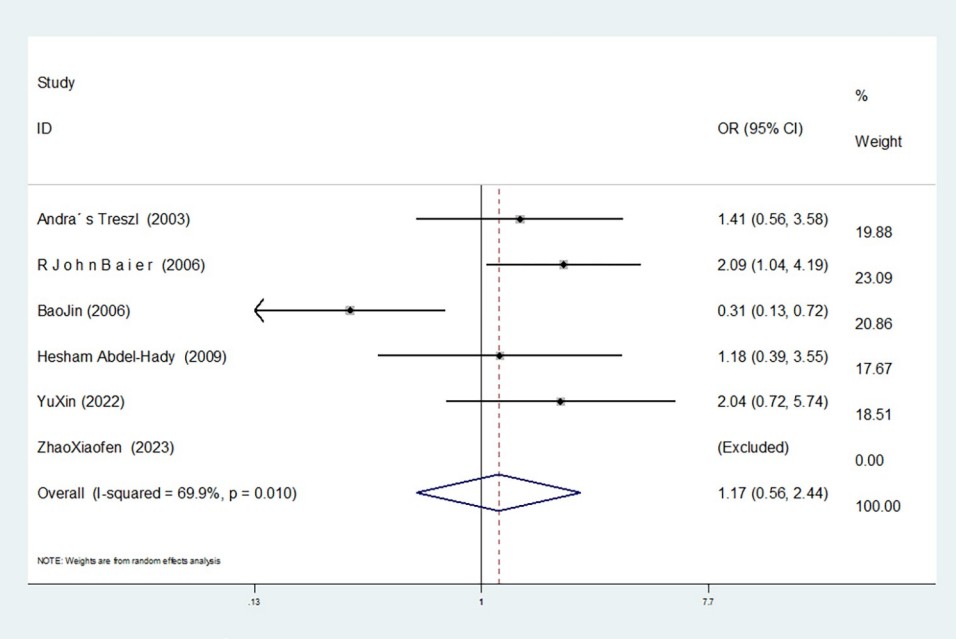

**Fig 31. TSA map of TNF-α recessive gene.**

indicating statistical significance. In the homozygous gene model ($P = 0.034$), $P$ was less than 0.05 and greater than 0.0083, indicating the possibility of false-positive results. However, further studies are required to test this hypothesis. The Bonferroni correction results for IL-8-rs4073 showed that $P$ was less than 0.0083 in both the recessive ($P = 0.003$) and the additive gene models ($P = 0.003$), indicating statistical significance. The Bonferroni

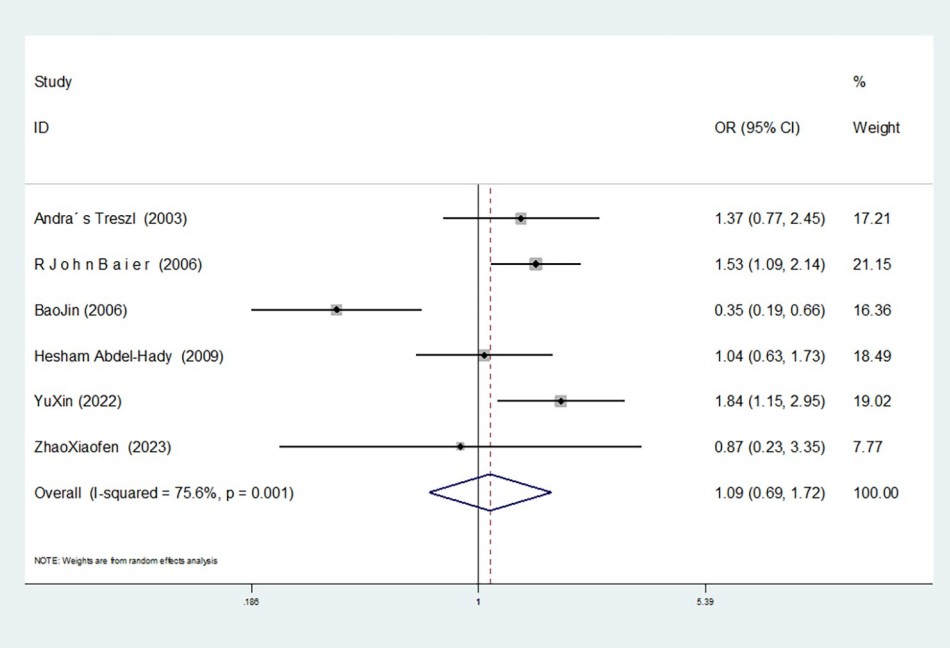

**Fig 32. TSA map of TNF-α dominant gene.**

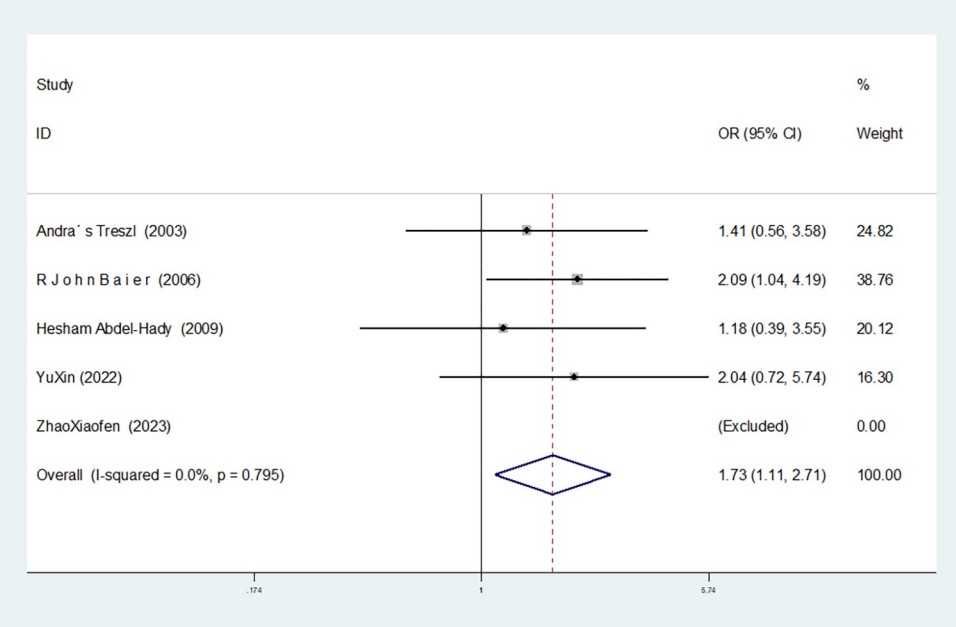

**Fig 33. TSA map of TNF-α heterozygous gene.**

correction results for IL-10-1082 G/A showed that *P* was less than 0.0083 in both the recessive gene model (*P* = 0.002) and the homozygous gene model (*P* = 0.002), indicating statistical significance. In the dominant gene model (*P* = 0.016), *P* was less than 0.05 and greater than 0.0083, indicating the possibility of false-positive results. However, further studies are required to test this hypothesis. The Bonferroni correction results of TNF- α– 308G/A

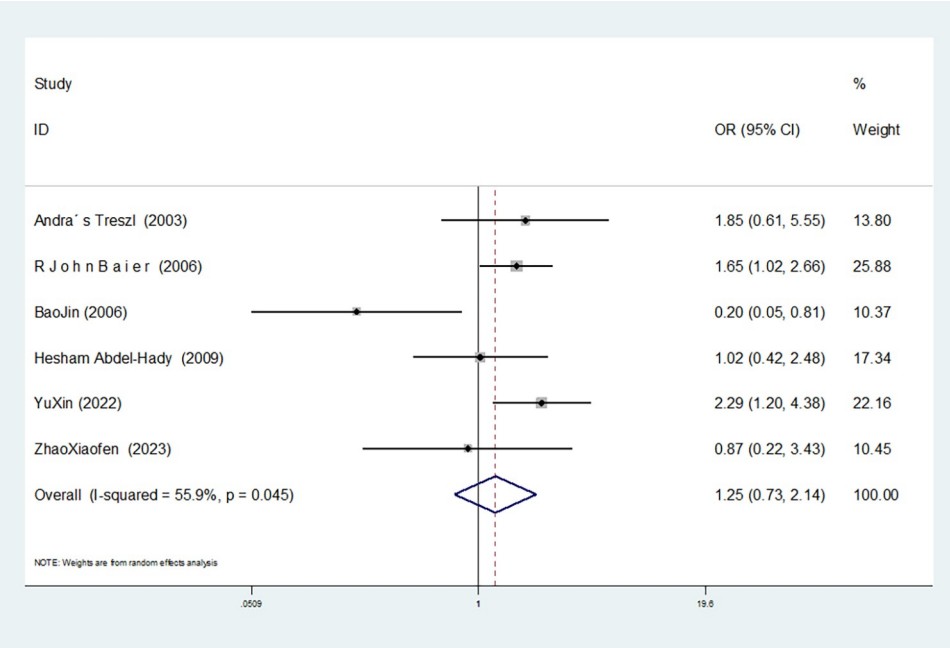

**Fig 34. TSA map of TNF-α homozygous gene.**

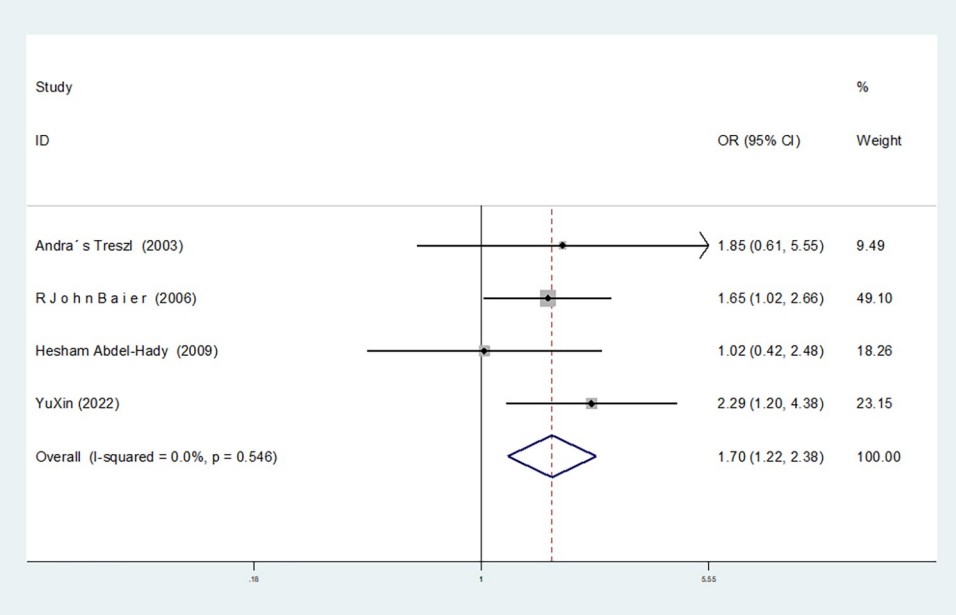

**Fig 35. TSA map of TNF-α additive gene.**

(rs1800629) showed that *P* was less than 0.05 and greater than 0.0083 in both the allele model (*P* = 0.016) and the recessive gene model (*P* = 0.009), indicating the possibility of false-positive results and further experiments are needed to verify.

1. False-positive reporting rate (FPRP) (results shown in Table 5): The critical value of FPRP was set to 0.2, with an OR of 1.5. For the prior probability π of H1, we divide it into three categories: low (≈0.001), medium (≈0.01), and high (≈0.1) [65–67], and calculate them using the table proposed by Wacholder et al [67]. The FPRP results of IL-1β rs1143643 show that in the allele model, PRPP cannot be calculated due to the small *P*-value, and it cannot be determined whether it is a false-positive result at this time. In the heterozygous gene model, π≈0.1, the FPRP is 0.116, which is lower than the preset value of 0.2. At π≈0.01 and π≈0.001, the FPRP is 0.591 and 0.936, respectively, higher than the preset value of 0.2, indicating statistical significance. However, additional experiments should be included for verification purposes. The FPRP results of IL-6-174G/C showed that in the recessive gene model, at π≈0.1, the FPRP was 0.104, lower than the preset value of 0.2. At π≈0.01 and π≈0.001, the FPRP was 0.560 and 0.928, respectively, higher than the preset value of 0.2, indicating statistical significance. However, additional experiments should be included for verification purposes. In the homozygous gene model, the FPRP at π≈0.1, π≈0.01, and π≈0.001 were 0.359, 0.861, and 0.984, respectively, which were higher than the preset values of 0.2, indicating the possibility of false-positive results and further experiments are needed to verify. The FPRP results of IL-8-rs4073 showed that in the recessive gene model, the FPRP values were 0.263, 0.797, and 0.975 at π≈0.1, π≈0.01, and π≈0.001, respectively, which were higher than the preset values of 0.2, indicating the possibility of false-positive results and further experiments are needed to verify. In the additive gene model, at π≈0.1, FPRP = 0.190, which is lower than the preset value of 0.2. At π≈0.01 and π≈0.001, FPRP is 0.721 and 0.963, respectively, higher than the preset value of 0.2,

**Table 4. Bonferroni calibration results.**

| IL-1βrs1143643 | | | | | |
|---|---|---|---|---|---|
| Model | Number of studies | Meta-analysis results | | Predetermined | Bonferroni correction |
| | | OR(95%CI) | $P$ | $P = 0.05$ | $P = 0.0083$ |
| Allelic model (T vs C) | 2 | 0.224 (0.168–0.299) | 0.000 | Yes | Yes |
| Dominant gene model (TT+CT vs CC) | 2 | 2.785 (0.963–8.052) | 0.059 | No | No |
| Recessive gene model (TT vs CT+CC) | 3 | 1.567 (0.934–2.631) | 0.089 | No | No |
| Heterozygous gene model (TC vs CC) | 2 | 4.251 (2.226–8.119) | 0.000 | Yes | Yes |
| Homozygous gene model (TT vs CC) | 3 | 2.020 (1.122–3.639) | 0.019 | Yes | No |
| Additive gene model (TT+CC vs TC) | 4 | 0.904 (0.377–2.171) | 0.822 | No | No |
| IL6-174G/C | | | | | |
| Model | Number of studies | Meta-analysis results | | Predetermined | Bonferroni correction |
| | | OR(95%CI) | $P$ | $P = 0.05$ | $P = 0.0083$ |
| Allelic model (C vs G) | 8 | 1.140 (0.974–1.335) | 0.101 | No | No |
| Dominant gene model (CC+GC vs GG) | 9 | 1.012 (0.689–1.488) | 0.950 | No | No |
| Recessive gene model (CC vs GC+GG) | 8 | 1.591 (1.154–2.194) | 0.005 | Yes | Yes |
| Heterozygous gene model (CG vs GG) | 6 | 1.021 (0.692–1.508) | 0.916 | No | No |
| Homozygous gene model (CC vs GG) | 8 | 1.471 (1.030–2.101) | 0.034 | Yes | No |
| Additive gene model (CC+GG vs CG) | 6 | 1.059 (0.835–1.343) | 0.635 | No | No |
| IL-8rs4073 | | | | | |
| Model | Number of studies | Meta-analysis results | | Predetermined | Bonferroni correction |
| | | OR(95%CI) | $P$ | $P = 0.05$ | $P = 0.0083$ |
| Allelic model (T vs A) | 2 | 0.756 (0.549–1.041) | 0.087 | No | No |
| Dominant gene model (TT+AT vs AA) | 2 | 1.112 (0.633–1.951) | 0.713 | No | No |
| Recessive gene model (TT vs AT+AA) | 2 | 0.467 (0.280–0.777) | 0.003 | Yes | Yes |
| Heterozygous gene model (TA vs AA) | 2 | 1.548 (0.855–2.806) | 0.149 | No | No |
| Homozygous gene model (TT vs AA) | 2 | 0.639 (0.328–1.243) | 0.187 | No | No |
| Additive gene model (TT+AA vs TA) | 2 | 0.497 (0.315–0.785) | 0.003 | Yes | Yes |
| IL-10–1082 G/A | | | | | |

*(Continued)*

**Table 4.** (Continued）

| Model | Number of studies | Meta-analysis results | | Predetermined | Bonferroni correction |
|---|---|---|---|---|---|
| | | OR(95%CI) | P | P = 0.05 | P = 0.0083 |
| Allelic model (A vs G) | 6 | 1.090 (0.691–1.719) | 0.712 | No | No |
| Dominant gene model (AA+GA vs GG) | 5 | 1.731 (1.108–2.705) | 0.016 | Yes | No |
| Recessive gene model (AA vs GA+GG) | 4 | 1.702 (1.218–2.377) | 0.002 | Yes | Yes |
| Heterozygous gene model (AG vs GG) | 6 | 1.110 (0.739–1.666) | 0.616 | No | No |
| Homozygous gene model (AA vs GG) | 5 | 2.282 (1.366–3.812) | 0.002 | Yes | Yes |
| Additive gene model (AA+GG vs AG) | 6 | 1.240 (0.934–1.645) | 0.136 | No | No |
| **TNF-α−308G/A(rs1800629)** | | | | | |
| Model | Number of studies | Meta-analysis results | | Predetermined | Bonferroni correction |
| | | OR(95%CI) | P | P = 0.05 | P = 0.0083 |
| Allelic model (A vs G) | 6 | 1.257 (1.044–1.513) | 0.016 | Yes | No |
| Dominant gene model (AA+GA vs GG) | 8 | 1.023 (0.803–1.303) | 0.854 | No | No |
| Recessive gene model (AA vs GA+GG) | 7 | 1.913 (1.179–3.104) | 0.009 | Yes | No |
| Heterozygous gene model (AG vs GG) | 7 | 0.948 (0.797–1.128) | 0.546 | No | No |
| Homozygous gene model (AA vs GG) | 8 | 1.341 (0.881–2.043) | 0.171 | No | No |
| Additive gene model (AA+GG vs AG) | 7 | 1.079 (0.908–1.282) | 0.386 | No | No |

Note: Yes represents statistically significant results, No represents non statistically significant results.

indicating statistical significance. However, additional experiments should be included for verification purposes. The FPRP results of IL-10-1082 G/A showed that in the dominant gene model, the FPRP values were 0.352, 0.857, and 0.984 at $\pi\approx0.1$, $\pi\approx0.01$, and $\pi\approx0.001$, respectively, which were higher than the preset values of 0.2, indicating the possibility of false-positive results and further experiments are needed to verify. In the recessive gene model, at $\pi\approx0.1$, FPRP = 0.066, which is lower than the preset value of 0.2. At $\pi\approx0.01$ and $\pi\approx0.001$, FPRP is 0.438 and 0.887, respectively, higher than the preset value of 0.2, indicating statistical significance. However, additional experiments should be included for verification purpose. In the homozygous gene model, the FPRP values were 0.574, 0.937, and 0.993 at $\pi\approx0.1$, $\pi\approx0.01$, and $\pi\approx0.001$, respectively, all higher than the preset values of 0.2, indicating the possibility of false-positive results and further experiments are needed to verify.

**Table 5. FPRP results.**

| IL-1βrs1143643 | | | | | | |
|---|---|---|---|---|---|---|
| Model | Number of studies | Meta-analysis results | | Prior probability | | |
| | | OR(95%CI) | P | 0.1 | 0.01 | 0.001 |
| Allelic model (T vs C) | 2 | 0.224 (0.168–0.299) | 0.000 | invalid | invalid | invalid |
| Dominant gene model (TT+CT vs CC) | 2 | 2.785 (0.963–8.052) | 0.059 | 0.123 | 0.606 | 0.940 |
| Recessive gene model (TT vs CT+CC) | 3 | 1.567 (0.934–2.631) | 0.089 | 0.649 | 0.953 | 0.995 |
| Heterozygous gene model (TC vs CC) | 2 | 4.251 (2.226–8.119) | 0.000 | 0.116 | 0.591 | 0.936 |
| Homozygous gene model (TT vs CC) | 3 | 2.020 (1.122–3.639) | 0.019 | 0.518 | 0.922 | 0.992 |
| Additive gene model (TT+CC vs TC) | 4 | 0.904 (0.377–2.171) | 0.822 | 0.889 | 0.989 | 0.999 |

Note: Display invalid due to *P* value being too small.

| IL6-174G/C | | | | | | |
|---|---|---|---|---|---|---|
| Model | Number of studies | Meta-analysis results | | Prior probability | | |
| | | OR(95%CI) | P | 0.1 | 0.01 | 0.001 |
| Allelic model (C vs G) | 8 | 1.140 (0.974–1.335) | 0.101 | 0.483 | 0.911 | 0.990 |
| Dominant gene model (CC+GC vs GG) | 9 | 1.012 (0.689–1.488) | 0.950 | 0.896 | 0.990 | 0.999 |
| Recessive gene model (CC vs GC+GG) | 8 | 1.591 (1.154–2.194) | 0.005 | 0.104 | 0.560 | 0.928 |
| Heterozygous gene model (CG vs GG) | 6 | 1.021 (0.692–1.508) | 0.916 | 0.894 | 0.989 | 0.999 |
| Homozygous gene model (CC vs GG) | 8 | 1.471 (1.030–2.101) | 0.034 | 0.359 | 0.861 | 0.984 |
| Additive gene model (CC+GG vs CG) | 6 | 1.059 (0.835–1.343) | 0.635 | 0.852 | 0.984 | 0.998 |

| IL-8rs4073 | | | | | | |
|---|---|---|---|---|---|---|
| Model | Number of studies | Meta-analysis results | | Prior probability | | |
| | | OR(95%CI) | P | 0.1 | 0.01 | 0.001 |
| Allelic model (T vs A) | 2 | 0.756 (0.549–1.041) | 0.087 | 0.500 | 0.917 | 0.991 |
| Dominant gene model (TT+AT vs AA) | 2 | 1.112 (0.633–1.951) | 0.713 | 0.883 | 0.988 | 0.999 |
| Recessive gene model (TT vs AT+AA) | 2 | 0.467 (0.280–0.777) | 0.003 | 0.263 | 0.797 | 0.975 |

(*Continued*)

**Table 5.** (Continued)

| Model | Number of studies | OR(95%CI) | P | 0.1 | 0.01 | 0.001 |
|---|---|---|---|---|---|---|
| Heterozygous gene model (TA vs AA) | 2 | 1.548 (0.855–2.806) | 0.149 | 0.746 | 0.970 | 0.997 |
| Homozygous gene model (TT vs AA) | 2 | 0.639 (0.328–1.243) | 0.187 | 0.789 | 0.976 | 0.998 |
| Additive gene model (TT+AA vs TA) | 2 | 0.497 (0.315–0.785) | 0.003 | 0.190 | 0.721 | 0.963 |

**IL-10−1082 G/A**

| Model | Number of studies | Meta-analysis results | | Prior probability | | |
|---|---|---|---|---|---|---|
| | | OR(95%CI) | P | 0.1 | 0.01 | 0.001 |
| Allelic model (A vs G) | 6 | 1.090 (0.691–1.719) | 0.712 | 0.328 | 0.843 | 0.982 |
| Dominant gene model (AA+GA vs GG) | 5 | 1.731 (1.108–2.705) | 0.016 | 0.352 | 0.857 | 0.984 |
| Recessive gene model (AA vs GA+GG) | 4 | 1.702 (1.218–2.377) | 0.002 | 0.066 | 0.438 | 0.887 |
| Heterozygous gene model (AG vs GG) | 6 | 1.110 (0.739–1.666) | 0.616 | 0.856 | 0.985 | 0.998 |
| Homozygous gene model (AA vs GG) | 5 | 2.282 (1.366–3.812) | 0.002 | 0.211 | 0.747 | 0.967 |
| Additive gene model (AA+GG vs AG) | 6 | 1.240 (0.934–1.645) | 0.136 | 0.574 | 0.937 | 0.993 |

**TNF-α−308G/A(rs1800629)**

| Model | Number of studies | Meta-analysis results | | Prior probability | | |
|---|---|---|---|---|---|---|
| | | OR(95%CI) | P | 0.1 | 0.01 | 0.001 |
| Allelic model (A vs G) | 6 | 1.257 (1.044–1.513) | 0.016 | 0.126 | 0.614 | 0.941 |
| Dominant gene model (AA+GA vs GG) | 8 | 1.023 (0.803–1.303) | 0.854 | 0.872 | 0.987 | 0.999 |
| Recessive gene model (AA vs GA+GG) | 7 | 1.913 (1.179–3.104) | 0.009 | 0.323 | 0.840 | 0.981 |
| Heterozygous gene model (AG vs GG) | 7 | 0.948 (0.797–1.128) | 0.546 | 0.831 | 0.982 | 0.998 |
| Homozygous gene model (AA vs GG) | 8 | 1.341 (0.881–2.043) | 0.171 | 0.689 | 0.961 | 0.996 |
| Additive gene model (AA+GG vs AG) | 7 | 1.079 (0.908–1.282) | 0.386 | 0.777 | 0.975 | 0.997 |

## 4. Discussion

Neonatal sepsis is a serious infectious disease in the neonatal care unit that has a significant impact on the survival rate of newborns [26]. In the neonatal period, the infant's immune

system develops very slowly, is relatively naive, has low immune ability [68], and is extremely susceptible to pathogens. The onset of neonatal sepsis is rapid and once infection occurs, it is extremely challenging to manage [69]. Neonatal sepsis is characterized by a systemic inflammatory response. Genetic mutations that occur during the inflammatory response may play important roles in the occurrence, development and outcome of neonatal sepsis [70–72]. The mechanism of this gene variation may enable us to discover new diagnostic and treatment methods for neonatal sepsis and further detect, diagnose and treat neonatal sepsis early, thereby improving prognosis.

Cytokines are proteins with a variety of functions that are generally synthesized and secreted by immune cells such as monocytes and macrophages. There are many cytokines, such as interleukin, interferon, tumor necrosis factor, colony-stimulating factor, growth factor and transforming growth factor β. Cytokines are involved in many biological functions. Inflammatory cytokines play an important role in the inflammatory response. Inflammatory cytokines have different roles, and according to their roles, they are divided into pro-inflammatory cytokines (such as IL-1β, IL-6, IL-8, TNF-α, etc.) and anti-inflammatory cytokines (such as IL-4, IL-10, etc.) [73]. Pro-inflammatory cytokines can promote the inflammatory response and induce the production of more inflammatory cytokines, thus helping the body to complete the inflammatory immune process. Anti-inflammatory cytokines, as their name suggests, work to suppress excessive inflammatory responses. Both pro-inflammatory and anti-inflammatory cytokines have significant prognostic benefits in patients with systemic inflammation and sepsis.

Advancements in human genomics have shed light on the relationship between individual variations and gene polymorphism [74, 75]. Two or more alleles in a gene locus on the human chromosome are called polymorphisms [76, 77]. Gene polymorphisms can affect susceptibility to disease, diversity of clinical symptoms and differences in therapeutic effects during treatment [78]. Both single-nucleotide and high-repeat mutations can lead to genetic polymorphisms, with the single-nucleotide mutation being the most common, and includes single-base insertion, loss, and replacement [77]. Studies have shown that the pathogenicity of some diseases, such as septicemia and periodontitis, is relatively similar to that of single nucleotide polymorphisms, presenting novel avenue for the early detection, diagnosis, treatment and prognosis of these diseases. Thus, this paper delves into gene polymorphism of cytokines to elucidate their relationship with neonatal sepsis, in order to develop tailored diagnostic and treatment plans for septic diseases with high mortality in the neonatal period.

## 4.1 Interleukin-1 (IL-1) gene polymorphism

IL-1 is a peptide growth factor that can be secreted by various cells, such as monocytes and dendritic cells. The gene encoded by IL-1 is located on human chromosome 2q13-14 [77]. It has a powerful function and plays a very important role in the pathogenesis of sepsis by participating in the immune response, inflammation, fever, protein synthesis and other processes in the acute phase. IL-1 is divided into IL-1α, IL-1β and IL-1ra [76]. Although IL-1α and IL-1β have many similarities, they also have great differences, particularly in gene expression, mRNA stability, translation, processing and secretion regulation. IL-1α is mainly a mediator for regulating local inflammation and intracellular activities, while IL-1β, upon release by cells, acts as a systemic sex hormone-like mediator, playing an indispensable role in immune, inflammatory response, fever and other processes. When an imbalance occurs, several pathological conditions occur such as sepsis, rheumatoid arthritis, inflammatory bowel disease, leukemia, diabetes and atherosclerosis. IL-1ra has an antagonistic effect only on the interleukin-1 family.

The genetic polymorphism of the IL-1α gene involves a substitution from cytosine (C) to thymine (T) at position 889 in the promoter region, along with a small satellite DNA containing 46 base pairs in exon 6. However, due to limited research on the relationship between IL-1α genetic polymorphism and sepsis, it was excluded from consideration.

Currently, it has been found that there is a genetic polymorphism site in the second intron of IL-1ra, similar to exon 6 in IL-1α, with both consisting of small satellite DNA composed of 86 base pairs. There was a mutation from thymine (T) to cytosine (C) at position 8006 in exon 2. A study by scholars [79] suggests that the IL-1Ra allele A2 may increase the risk of systemic infection in the population. The relevant sample size collected in this study was insufficient and was therefore excluded.

IL-1β Known as a "superhero warrior", comprises six introns and seven exons. It is predominantly expressed in the hypothalamus, cerebral cortex and hippocampus. They are the mainly derived from neurons, microglia and astrocytes in the brain. Additionally, IL-1β can induce the production of inflammatory factors such as IL-6 and TNF-α [80], play the role of immune response, immune regulation and endocrine, and can timely warn the abnormal metabolic activities of the body. Currently discovered IL-1β genetic polymorphisms are mostly biallelic polymorphisms, including cytosine (C) to thymine (T) substitution at position 31 of the promoter and exon regions, thymine (T) to cytosine (C) substitution at position 511, guanine (G) to cytosine (C) substitution at position 1470, cytosine (C) to thymine (T) substitution at position 3953, and cytosine (C) to thymine (T) substitution at position 5887 permutation and a polymorphic restriction site with Taq I on exon 5 [81]. These gene mutation sites are associated with the occurrence, development, diagnosis and prognosis of many infectious diseases such as periodontitis, meningococcal meningitis and enteritis [82]. Some studies have found that IL-1β the mutation of thymine (T) to cytosine (C) at site 511 in the promoter region is significantly associated with the onset of many inflammatory diseases [42], but due to limited relevant studies, this study was excluded from our analysis. Mutations in cytosine (C) to thymine (T) at site 3954 of IL-1β exon 5 may be associated with sepsis, but there are also contrary opinions [44]. While Zhao Xiaofen et al. [35] believed that IL-1β gene loci rs1143627(C/T) and rs1143643(C/T) may not be associated with neonatal sepsis. Susanna Esposito et al.[33] posits that IL-1β gene rs1143643 was significantly associated with the overall risk of sepsis. This article mainly summarizes and statistically analyzes the controversial clinical research on the association between IL-1βrs1143643 polymorphism and neonatal sepsis [32–35]. Finally, it was found that the OR value for the IL-1β rs1143643 polymorphism in the allele model was statistically significant ($P$ = 0.000). This suggests that newborns with the T allele genotype have a lower risk of sepsis (OR = 0.224, 95%w CI: 0.168–0.299). Similarly, the OR values merged in the heterozygous gene model were statistically significant ($P$ = 0.000), indicating a significant increase in the risk of neonatal sepsis with the TC genotype (OR = 4.251, 95% CI: 2.226–8.119). Additionally, the OR values merged in the homozygous gene model were statistically significant ($P$ = 0.019), indicating a significant increase in the risk of sepsis in newborns with TT genotype (OR = 2.020, 95% CI: 1.122–3.639).

### 4.2 Interleukin-6 (IL-6) gene polymorphism

IL-6 is a pro-inflammatory cytokine generally expressed in adipocytes. It is mainly involved in the inflammatory response, hematopoietic processes, stimulation and activation of B cell proliferation [73]. Both TNF and IL-1 can induce the production and secretion of IL-6 [73], with its concentration rising most rapidly during inflammation. Moreover, its plasma concentration is closely related to sepsis mortality and can be a reliable predictor of sepsis prognosis [82].

IL-6 is located on human chromosome 7 and consists of four introns and five exons. Currently, it has been found that the genetic polymorphism of IL-6 involves the substitution of guanine (G) to adenine (A) at position 597 in the promoter region, cytosine (C) to guanine (G) at position 572, guanine (G) to cytosine (C) at position 174, and AnTn at position 137 [83]. Among them, site 174 was the most studied, and the C allele promoted reduced IL-6 production. PETER AHRENS et al.[37] found a significant association between the homozygous IL6-174G genotype and the development of sepsis in low-birth weight infants. There are also studies [84] showing that gene polymorphism of the IL-6 locus has a greater impact on the prognosis of sepsis but has little impact on the pathogenesis of sepsis. Therefore, this study summarized and analyzed the literature on the correlation between the IL-6 174G polymorphism and neonatal sepsis [34, 36–43], and finally found that the OR value of the IL-6-174G/C polymorphism merged in the recessive gene model was statistically significant ($P = 0.005$), indicating a significant increase in the risk of neonatal sepsis with the CC genotype (OR = 1.591, 95% CI: 1.154–2.194). The OR values merged in the homozygous gene model were statistically significant ($P = 0.034$), indicating a significant increase in the risk of sepsis in newborns with CC genotype (OR = 1.471, 95% CI: 1.030–2.101).

### 4.3 Interleukin-8 (IL-8) gene polymorphism

IL-8 belongs to the chemokine subfamily, also known as CXCL8, which regulates neutrophil chemotaxis [85, 86]. Mononuclear macrophages are the main cells that produce IL-8, while fibroblasts, epithelial cells, and endothelial cells produce IL-8 under appropriate stimulation. IL-8 regulates both acute and chronic inflammation and is involved in the occurrence and development of various diseases such as peptic ulcers, pancreatitis, malaria, gastric cancer and breast cancer [86]. Studies have found that IL-8 acts as an angiogenic factor, affects the occurrence and development of atherosclerosis, and plays an early warning role in cardiovascular diseases. In patients with sepsis after severe trauma or burns, IL-8 is elevated be used to diagnose early neonatal sepsis with relatively high accuracy.

In recent years, researchers have found that some sites in the promoter region of IL-8 may be related to sepsis. IL-8 is located on the human chromosome 4 and contains four exons and three introns [86, 87]. Currently, genetic polymorphisms of IL-8 have been found in the substitution of thymine (T) to cytosine (C) at position rs2227306 in intron 1, thymine (T) to adenine (A) at position rs1126647 in exon 4, and thymine (T) to adenine (A) at position rs4073 in the 5' flanking region [87]. The results show that the rs4073 and rs2227306 polymorphisms of the 5' flanking region and intron 1 of IL-8 are mainly involved in the transcription process of IL-8. To date, no research has been conducted on the relationship between genetic polymorphisms at the rs2227306C/T and rs1126647A/T loci in IL-8 and sepsis. More research has been conducted on the genetic variations at the rs4073 locus. Chinese scholars believe that genetic polymorphisms of the IL-8rs4073A/T locus are related to post-traumatic sepsis and organ failure. However, whether there is a correlation with neonatal sepsis remains a concern. Susanna Esposito et al. [33] suggested that the rs4073(IL-8 gene) genotype AT is associated with a significantly increased risk of severe sepsis, and Zhao Xiaofen et al. [44] suggested that the TT genotype at the rs4073 locus of IL-8 might be associated with susceptibility to sepsis in full-term newborns. This study summarizes and analyzes the literature on the association between the IL-8rs4073 polymorphism and neonatal sepsis [36, 44], and found that the OR value of the IL-8-rs4073 polymorphism merged in the recessive gene model was statistically significant ($P = 0.003$), indicating that the risk of neonatal sepsis was significantly reduced in infants with the TT genotype (OR = 0.467, 95% CI: 0.280–0.777). The OR values merged in the additive

gene model were statistically significant ($P$ = 0.003), indicating a significant reduction in the risk of sepsis in newborns with the TT+AA genotypes (OR = 0.497, 95% CI: 0.315–0.785).

## 4.4 Interleukin-10 (IL-10) gene polymorphism

IL-10 is an anti-inflammatory cytokine that plays a powerful anti-inflammatory and immuno-suppressive role in the body. It can correct excessive inflammatory responses and ensure the stability of the body's internal environment [20]. IL-10 is produced by lymphocytes in vivo, with mononuclear macrophages being the main production sites. The IL-10 is located on the human chromosome 1 and consists of five exons [20]. It has been found that an inflammatory response that is too low in the expression of anti-inflammatory cytokines has a significant impact on the onset of sepsis [88]; therefore, IL-10 can regulate the occurrence, development and prognosis of sepsis. Genetic variation factors affect the amount of IL-10 secreted by cells in the body, so as to further regulate sepsis.

Three base replacements have been found in the promoter region of the IL-10 gene, namely mutations of guanine (G) at 1082 to adenine (A), cytosine (C) at 819 to thymine (T), and cytosine (C) at 592 to adenine (A) [81, 89]. Genetic polymorphism sites were also found in the intron region, with adenine (A) mutated to guanine (G) at 2849, guanine (G) mutated to cytosine (C) at 1593, guanine (G) mutated to adenine (A) at 1388, adenine (A) mutated to guanine (G) at 1147, and adenine (A) mutated to guanine (G) at 2650 [81]. Currently, the three gene loci in the promoter region are the most widely studied. Some studies [90] indicated that the gene polymorphisms at loci 592 and 819 of the IL-10 promoter region had no effect on sepsis; therefore, they were excluded from this study. An adenine (A) mutation to guanine(G) at site 1082 of the IL-10 promoter region reduces the content of intracellular IL-10, thus further affecting the occurrence and development of sepsis. Domestic scholars [91] have found that a single nucleotide polymorphism in the IL-10-1082 gene is closely related to the tendency to develop sepsis. Andras Treszl et al. [36] reported that the polymorphism of the IL-10-1082 gene locus was not associated with neonatal sepsis, whereas Abu-Maziad et al. [92] found that the IL-10-1082GG genotype was significantly associated with a reduced risk of sepsis in very low birth weight infants and preterm infants. To further understand the relationship between polymorphism of IL-10-1082 gene loci and neonatal sepsis, we summarized domestic and foreign studies on this topic [38, 43, 44–47], and found that the OR value of the IL-10-1082 G/A polymorphism merged in the dominant gene model was statistically significant ($P$ = 0.016), indicating a significant increase in the risk of neonatal sepsis with the AA+GA genotype (OR = 1.731, 95% CI: 1.108–2.705). The OR values merged in the recessive gene model were statistically significant ($P$ = 0.002), indicating a significant increase in the risk of sepsis in newborns with AA genotype (OR = 1.702, 95% CI: 1.218–2.377). The OR values merged in the homozygous gene model were statistically significant ($P$ = 0.002), indicating a significant increase in the risk of sepsis in newborns with AA genotype (OR = 2.282, 95% CI: 1.366–3.812).

## 4.5 Tumor growth factor-α (TNF-α) gene polymorphism

TNF is the earliest cytokine released by the body and is a candidate gene for serious infectious diseases. It is located on human chromosome 6, including TNF A and TNF B, encoding TNF-α and TNF-β respectively [89]. TNF-α and TNF-β have similar functions, both of which can stimulate and regulate inflammatory response, anti-tumor, anti-virus, etc., and can also induce the generation of new blood vessels. Although, TNF-α is generally secreted by neutrophils, activated lymphocytes and macrophages and plays a role mainly through the two receptors, TNF-R1 and TNF-R2, which are pro-inflammatory cytokines, TNF-β involves the production

of activated lymphocytes. Currently, TNF-α is the most studied and deeply understood cytokine in the research related to sepsis. This is a result of its direct impact on the occurrence and development of sepsis.

Currently, it has been found that some regions upstream of the promoter are mainly TNF-α. The locations where single nucleotide polymorphisms occur [89], including thymine (T) mutation to cytosine (C) at 1031, cytosine (C) mutation to adenine (A) at 863, cytosine (C) mutation to adenine (A) at 857, cytosine (C) mutation to thymine (T) at 850, guanine (G) mutation to cytosine (C) at 419, and guanine (G) mutation to adenine (A) at 376, mutation of adenine (A) to guanine (G) at position 308 and guanine (G) to adenine (A) at position 238 [81]. Presently, the most extensively studied mutation in this region is at site 308. This mutation is biallelic, with the presence of G being more common known as TNF-α1, and presence of A being relatively rare referred to as TNF-α2 [82]. In recent years, it has been found that TNF-α2 has an effect on the activity of TNF-α promoter and can regulate the production of TNF-α. Tatjana Varljen et al. [8] found that TNF-α-308A allele may be a risk factor for early-onset neonatal sepsis, while Lakshmi Srinivasan et al. [28] did not find an association between tumor necrosis factor-308g /A polymorphism and sepsis. Therefore, we summarized relevant literature at home and abroad [34, 36, 41, 48–52] and found that the OR value of TNF-α-308G/A (rs1800629) polymorphism merged in the allele model was statistically significant ($P$ = 0.016), indicating a significant increase in the risk of neonatal sepsis with the A allele (OR = 1.257, 95% CI: 1.044–1.513). The OR value merged in the recessive gene model was statistically significant ($P$ = 0.009), indicating a significant increase in the risk of sepsis in newborns with the AA genotype (OR = 1.913, 95% CI: 1.179–3.104).

In conclusion, this study described the relationship between candidate gene polymorphisms and neonatal sepsis. Given the varying sample sizes related to each cytokine, we summarized and analyzed them to enhance the reliability of the study. Through individual exclusion of each study individually and subsequent sensitivity analysis, the results showed that the results were relatively stable. The results of Begg's and Egger's tests indicated no significant publication bias. The mata-analysis results of IL-1βrs1143643 polymorphism across six genetic models suggest that newborns with T alleles have a lower risk of sepsis. Combined with the TSA, Bonferroni correction, and FPRP results, this result is highly credible. The risk of neonatal sepsis was significantly increased with the TC genotype. Combined with the TSA, Bonferroni correction, and FPRP results, this result may be a false-positive, and further experiments are required to verify this. The risk of sepsis in newborns with the TT genotype was significantly increased. Combined with the TSA, Bonferroni correction, and FPRP results, this result may be a false-positive, and further experiments are required to verify this. Similar uncertainties exist regarding the association of neonatal sepsis risk with the CC genotype of the IL-6-174G/C polymorphism and the TT, TT+AA genotype of the IL-8-rs4073 polymorphism. Furthermore, the meta-analysis results suggest a significantly increased risk of neonatal sepsis with the AA+GA, AA genotype of the IL-10-1082 G/A polymorphism in six genetic models and the A allele, AA genotype of TNF-α-308G/A (rs1800629) polymorphism. However, further experiments are required to verify these findings.

The results of this study differ from those of previous studies on adult sepsis, indicating that there are differences in the growth and development of different individuals in the process of gene expression. Genetic mutations can affect newborns, but the magnitude and quality of the effects differ among adults, children, and newborns of different gestational ages and birth weights. This highlights the need for high-quality genome-wide association studies on neonatal sepsis. The strengths of this study include the following: (1) strict inclusion criteria for the retrieved literature; (2) The existence of publication bias verified through Begg's and Egger's tests to avoid unnecessary biases; (3) Conducting sequential experimental analysis to verify the

validity of the results; (4) Evaluating the occurrence of false-positive results in our study using Bonferroni correction and the FPRP method. At the same time, our study also has some limitations, such as: (1)The heterogeneity of individual studies is significant, with significant differences in the distribution of countries and races in individual studies. Differences in patient sex and gestational age, as well as differences in the definition and classification of sepsis, all contribute to the occurrence of heterogeneity to a certain extent; (2) There are four items included in the study that do not conform to the Hardy Weinberg equilibrium, and one item is not mentioned and cannot be calculated; (3) Some studies have small sample sizes, making it difficult to elucidate the association between cytokine gene polymorphisms and neonatal sepsis. Therefore, the results of the summary analysis may be included in the study's bias impact; (4) When conducting TSA experiments, Bonferroni calibration, and FPRP, parameter settings may be non-standard and have strong subjectivity. The results obtained by different parameters vary greatly, and strict control of false-positives may increase the risk of false-negatives to a certain extent; (5) The study was limited to the polymorphism of individual gene loci in various cytokines, thus excluding other loci; (6) The subgroup analysis only analyzed the effects of race and HWE, without considering other factors such as gestational age and gender; (7) Only a few cytokine gene polymorphisms were investigated for their relationship with susceptibility to neonatal sepsis, and no other cytokines or biomarkers were studied, and the impact of the interaction between genes and genes, as well as between genes and the environment, on neonatal sepsis has not been studied.

Although the above reasons may have influenced the results, this genome-wide study on cytokines and neonatal sepsis can provides strong evidence for the molecular genetic pathogenesis of neonatal sepsis. A more in-depth study of this genetic variation and disease association is of great significance for the development of individualized clinical treatment plans. Currently, the management of sepsis still focuses on the prevention of primary and secondary environmental risk factors, but increasing research on gene polymorphisms will provide new ideas for the prevention and management of neonatal sepsis.

## Supporting information

**S1 File. Excluded articles.**
(DOC)

**S2 File. Checklist.**
(DOCX)

**S3 File. Search strategy.**
(DOC)

**S4 File. . Certificate of editing.**
(PDF)

## Acknowledgments

We thank our colleagues for missing full-text articles and their invaluable help to translate the articles. The authors received no specific funding for this work and declare that no competing interests exist.

## Author Contributions

**Conceptualization:** Ling Hao.

**Data curation:** Jiaojiao Liang, Yan Su, Na Wang, Xiaoyan Wang.

**Formal analysis:** Jiaojiao Liang.

**Writing – original draft:** Jiaojiao Liang, Yan Su.

**Writing – review & editing:** Ling Hao, Changjun Ren.

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
