## [Decision Letter · Decision Letter 0]

5 Feb 2024

PONE-D-23-36527A meta-analysis of the association between inflammatory cytokine polymorphism and neonatal sepsisPLOS ONE

Dear Dr. Hao,

Thank you for submitting your manuscript to PLOS ONE. After careful consideration, we feel that it has merit but does not fully meet PLOS ONE’s publication criteria as it currently stands. Therefore, we invite you to submit a revised version of the manuscript that addresses the points raised during the review process.

The manuscript evaluated the link between inflammatory cytokine polymorphisms and neonatal sepsis with a meta-analysis. The authors should revise the manuscript according to the reviewers' suggestions and should respond to the reviewers point by point. Besides the reviewers' comments, I have the following observations regarding this manuscript.

1. The introduction/ literature section should be elaborated to reflect all the variations of five included genes, including sepsis.

2. A sequential trial analysis should be performed to verify that the included studies are sufficient to rationalize the studies' outcomes.

3. Due to the studies of multiple variants in multiple models, Bonferroni correction of p-value should be performed.

4. False-positive report probability (FPRP)  should be included to evaluate the credibility and reliability of observed associations or effects.

5. Authors should clarify which variant/model increases the risk of sepsis and which decreases the risk.

6. The authors should minimize the grammatical errors. 

We look forward to receiving your revised manuscript.

Kind regards,

Mohammad Safiqul Islam, Ph.D

Academic Editor

PLOS ONE

2. PLOS requires an ORCID iD for the corresponding author in Editorial Manager on papers submitted after December 6th, 2016. Please ensure that you have an ORCID iD and that it is validated in Editorial Manager. To do this, go to ‘Update my Information’ (in the upper left-hand corner of the main menu), and click on the Fetch/Validate link next to the ORCID field. This will take you to the ORCID site and allow you to create a new iD or authenticate a pre-existing iD in Editorial Manager. Please see the following video for instructions on linking an ORCID iD to your Editorial Manager account: " ext-link-type="uri" xlink:type="simple">https://www.youtube.com/watch?v=_xcclfuvtxQ".

Additional Editor Comments:

The manuscript evaluated the link between inflammatory cytokine polymorphisms and neonatal sepsis with a meta-analysis. The authors should revise the manuscript according to the reviewers' suggestions and should respond to the reviewers point by point. Besides the reviewers' comments, I have the following observations regarding this manuscript.

1. The introduction/ literature section should be elaborated to reflect all the variations of five included genes, including sepsis.

2. A sequential trial analysis should be performed to verify that the included studies are sufficient to rationalize the studies' outcomes.

3. Due to the studies of multiple variants in multiple models, Bonferroni correction of p-value should be performed.

4. False-positive report probability (FPRP) should be included to evaluate the credibility and reliability of observed associations or effects.

5. Authors should clarify which variant/model increases the risk of sepsis and which decreases the risk.

6. The authors should minimize the grammatical errors.

Reviewers' comments:

Reviewer's Responses to Questions

**Comments to the Author**

1. Is the manuscript technically sound, and do the data support the conclusions?

Reviewer #1: Yes

Reviewer #2: Yes

2. Has the statistical analysis been performed appropriately and rigorously? 

Reviewer #1: Yes

Reviewer #2: Yes

3. Have the authors made all data underlying the findings in their manuscript fully available?

Reviewer #1: Yes

Reviewer #2: Yes

4. Is the manuscript presented in an intelligible fashion and written in standard English?

Reviewer #1: Yes

Reviewer #2: Yes

5. Review Comments to the Author

Reviewer #1: The manuscript on inflammatory cytokine polymorphism and neonatal sepsis has several areas for improvement:

1. The literature collection could be more comprehensive. Currently, the literature mainly focuses on the gene polymorphisms of several inflammatory factors. Whether the search scope could be expanded to cover more inflammatory factors and other biomarkers related to neonatal sepsis.

2. The statistical methods could be described in more details, for example, the statistical reasoning of Hardy-Weinberg equilibrium test, heterogeneity test, etc.

3. The results describing the associations between various gene polymorphisms and neonatal sepsis could be more clear and concrete, so that readers do not have to search for the information. It is suggested to present the statistical testing results of associations between gene polymorphisms and disease in a table form.

4. The limitations of this study could be more comprehensively described in the discussion section. Currently, only heterogeneity and sample size are discussed. Whether there are other biases that need to be explored.

5. The conclusion is a bit absolute. It is suggested to use words like "indicate", "imply" instead of "confirm", as this study cannot completely "confirm" the associations between certain genes and disease.

6. The format of references could be unified.

7. The excessive number of pictures is very inconvenient for readers.

8. The language context in the manuscript is less coherent and the writing is more mechanical, so it is recommended to have it revised by a professional with a foundation in English as a mother tongue.

Reviewer #2: Reviewer’s Comments

General

English grammar needs to be improved, though it is comprehensible but not upto the mark.

Title. The title correctly reflects the aim and scope of the research.

ABSTRACT

The abstract follows a logical format, outlining the background, objectives, methods, results, and conclusions. Searching multiple databases increases the likelihood of capturing relevant literature. Appropriate statistical tests are used to assess heterogeneity and publication bias. The abstract details the significant associations identified for different cytokines and genetic models.

The abstract acknowledges the limitations of small sample size in some studies.

i. While the abstract states associations between specific polymorphisms and neonatal sepsis, it could benefit from mentioning the direction of the effect (e.g., increased/decreased risk).

ii. Quantifying the strength of evidence using measures like P values and confidence intervals would strengthen the abstract.

iii. Briefly mentioning the potential clinical significance of the findings (e.g., for diagnosis or personalized medicine) could enhance the abstract's impact.

Collectively, this is a well-written abstract that effectively summarizes the main findings of the meta-analysis.

1. INTRODUCTION

The introduction section of the meta-analysis clearly states the importance of neonatal sepsis and its connection to inflammation and also highlights the role of cytokines in immune response and potential connection to sepsis susceptibility. It also mentions the inconsistency of previous research on cytokine polymorphisms and neonatal sepsis and also clearly describes the aim of the current study

i. Though the introduction section is good but very short and broad. It could be improved by broad-to-narrow manner, to a more specific statement about the study's hypothesis or key research question.

ii. Repetition. Sentences 59-60 and 63 could be combined to avoid repetition.

iii. In the reviewer opinion, the meta-analysis needs to be supported by more and recent reference (if available). References are only used once in the introduction, making it difficult to assess the strength of the claims made. Consider incorporating more specific citations and evidence to support the statements.

iv. Information on the specific polymorphisms explored in the study is missing. Mentioning which cytokine genes and polymorphisms will be investigated would provide better context.

v. The introduction primarily focuses on genetic susceptibility and doesn't mention other potential factors influencing the association between cytokines and neonatal sepsis (e.g., environmental factors, maternal health). Briefly acknowledging these complexities would provide a more nuanced picture of the research topic.

vi. Conclude the introduction section with a clear statement of the study's objectives and its potential contribution to the field.

2. Materials and methods

2.1Document retrieval

The terms provided seem good, but justification for their inclusion (e.g., pilot searches, previous reviews) would support the rationale of the search strategy.

Line 72. Please rephrase the sentence to. To ensure a comprehensive search for relevant articles, we combined medically relevant keywords with broader terms. Additionally, when encountering duplicate publications, we selected the one with the most complete data for inclusion in the study.

For the above statement, the reviewer would like to know “How is the "most complete data" chosen among repeated studies?”. Explain in one or two lines.

2.2Inclusion and exclusion criteria

The meta-analysis carefully selected relevant studies by focusing on specific designs, prioritizing newborn populations, ensuring data quality, and excluding redundant or irrelevant research. Excellent.

Limiting the study to Chinese and English might exclude valuable research conducted in other languages. Consider justifying this choice or exploring translation options.

Like Figure 1, almost all other figures are not visible and readable though they seem correct. The resolution could be resolved for clarity, for all figures.

2.3Literature screening

Please remove the following subtitle from the text. Figures legend should be below or above the relevant figure and could be with more detailed description so that the figure looks self-explanatory.

Line 88 Fig 1. Flow Chart of IL-1 Literature Screening.

Line 91 Fig 2. Flow Chart of IL-6 Literature Screening.

Line 95 Fig 3. Flow Chart of IL-8 Literature Screening.

Line 99 Fig 4. Flow Chart of IL-10 Literature Screening

Line 103 Fig 5. Flow Chart of TNF- α Literature Screening

Please follow the same instructions for other such relevant descriptions.

2.5Quality evaluation

Briefly justify the choice of NOS for this specific meta-analysis context.

2.6Statistical analysis

Detailed and comprehensive statistical data and points are given. Excellent

In the reviewer opinion, Begg's funnel plot is undoubtedly a very useful and required tool for such analyses, but not necessarily, just a suggestion that considering additional methods like Egger's test for publication bias would provide a more comprehensive evaluation, if possible, please include it in the study, otherwise not obligatory.

3. Results

Overall, the results described based on the methodology are genuine. Each section effectively presents the general characteristics of the included studies, enhancing the study's transparency and allowing readers to evaluate its representativeness.

4. Conclusion

Conclusions are presented in an appropriate fashion and are supported by the data.

Decision.

The reviewer is quite impressed by the efforts and hard work of the authors and congratulate them for such a detailed and attractive study.

The article can be accepted for publication after the authors addressing the issues raised by the reviewer.

6. PLOS authors have the option to publish the peer review history of their article (what does this mean?). If published, this will include your full peer review and any attached files.

Reviewer #1: No

Reviewer #2: **Yes: **Dr. Arshad Islam, PhD, Assistant Professor, Medical Teaching Institution, Lady Reading Hospital, Peshawar 25100, Pakistan

---

## [Author Response · Author response to Decision Letter 0]

20 Mar 2024

Dear academic editor and reviewers:

RE: PONE-D-23-36527

We thank you and the expert reviewers for examining and approving our manuscript, “A Meta-analysis of the Association between Inflammatory Cytokine Polymorphism and neonatal sepsis” by Liang Jiaojiao et al. In this study, we conducted the correlation between single nucleotide polymorphisms of inflammatory cytokines and neonatal sepsis.

Thank you for your decision and constructive comments on my manuscript. We have carefully revised the manuscript text based on the reviewers’ suggestions and provided a point-to-point response to the comments. We herewith submit the revised manuscript with changes, the red part has been revised according to your comments, and a “clean” copy for reexamination. Revision notes, point-to-point, are given as follows:

SUGGESTIONS FROM THE ACADEMIC EDITOR:

1. The introduction/ literature section should be elaborated to reflect all the variations of five included genes, including sepsis.

AUTHOR RESPONSE:

 We think this is an excellent suggestion. We have added these instructions in the “4.1 Interleukin-1 (IL-1) gene polymorphism, 4.2 Interleukin-6 (IL-6) gene polymorphism, 4.3 Interleukin-8 (IL-8) gene polymorphism, 4.4 Interleukin-10 (IL-10) gene polymorphism and 4.5 Tumor growth factor-α (TNF-α) gene polymorphism” section according to your suggestion (Pages 39-44, Lines 668-669, 671-673, 680-684, 705-706, 725-727, 749-754, 777-781).

SUGGESTIONS FROM THE ACADEMIC EDITOR:

2.A sequential trial analysis should be performed to verify that the included studies are sufficient to rationalize the studies' outcomes.

AUTHOR RESPONSE: 

We agree with the academic editor’s assessment. Accordingly, throughout the manuscript, we have added a sequential trial analysis in the “3.9 Sequential analysis and multiple test correction” section (Pages 27-31, Lines 483-560).

SUGGESTIONS FROM THE ACADEMIC EDITOR:

3.Due to the studies of multiple variants in multiple models, Bonferroni correction of p-value should be performed.

AUTHOR RESPONSE: 

We agree with the academic editor’s assessment. Accordingly, throughout the manuscript, we have added the Bonferroni correction of p-value in the “3.9 Sequential analysis and multiple test correction” section (Pages 31-34, Lines 571-595).

SUGGESTIONS FROM THE ACADEMIC EDITOR:

4.False-positive report probability (FPRP) should be included to evaluate the credibility and reliability of observed associations or effects.

AUTHOR RESPONSE: 

We agree with the academic editor’s assessment. Accordingly, throughout the manuscript, we have added the False-positive report probability (FPRP) in the “3.9 Sequential analysis and multiple test correction” section (Pages 34-37, Lines 596-627).

SUGGESTIONS FROM THE ACADEMIC EDITOR:

5. Authors should clarify which variant/model increases the risk of sepsis and which decreases the risk.

AUTHOR RESPONSE: 

We agree with the academic editor’s assessment. Accordingly, throughout the manuscript, we have re-written this part in the “3.5 The relationship between polymorphisms of various cytokines and neonatal sepsis” section (Pages 15-16, Lines 218-256).

SUGGESTIONS FROM THE ACADEMIC EDITOR:

6. The authors should minimize the grammatical errors.

AUTHOR RESPONSE:

Thanks for your suggestion. However, we do invite a friend of ours who is a native English speaker from the USA to help polish our article. These changes will not influence the content and framework of the paper. Here we did not list the changes but marked them in red in the revised paper. We appreciate for Academic Editor and Reviewers’ warm work earnestly and hope that the correction will meet with approval.

Reviewer #1

REVIEWER COMMENTS:

1.The literature collection could be more comprehensive. Currently, the literature mainly focuses on the gene polymorphisms of several inflammatory factors. Whether the search scope could be expanded to cover more inflammatory factors and other biomarkers related to neonatal sepsis.

AUTHOR RESPONSE:

Thank you for pointing this out. There are many biomarkers for neonatal sepsis, which can be classified into four types: classical, non-traditional, novel, and others. Classic biomarkers mainly include white blood cell count, absolute neutrophil count, platelet count, C-reactive protein, procalcitonin, etc. Although they have a certain role in the diagnosis of neonatal sepsis, their specificity and sensitivity are insufficient, and they are easily influenced by other factors. Therefore, we did not search for relevant articles. Nontraditional biomarkers generally refer to cytokines, which can play an important role in the early diagnosis of neonatal sepsis. However, there is no unified diagnostic threshold in clinical practice. Currently, clinical attention is paid to the relationship between cytokines and neonatal sepsis, so we will mainly focus on cytokines. New biomarkers such as cell surface antigens have lower specificity and sensitivity, while other biomarkers such as micro MRA and nucleic acid analysis may become new biomarkers, but further validation is needed. Your question is quite insightful and cutting-edge, and we will pay close attention to the research in this field in the future. 

REVIEWER COMMENTS:

2. The statistical methods could be described in more details, for example, the statistical reasoning of Hardy-Weinberg equilibrium test, heterogeneity test, etc.

AUTHOR RESPONSE:

As suggested by the reviewer, we have added these instructions in the “2.6 Statistical analysis” section according to your suggestion (Pages 6-7, Lines 136-149).

REVIEWER COMMENTS:

3. The results describing the associations between various gene polymorphisms and neonatal sepsis could be more clear and concrete, so that readers do not have to search for the information. It is suggested to present the statistical testing results of associations between gene polymorphisms and disease in a table form.

AUTHOR RESPONSE:

We agree with the reviewer’s assessment. We have revised this part in the “3.4 Heterogeneity assessment (see Table 2) and 3.5 The relationship between polymorphisms of various cytokines and neonatal sepsis (see Table 2 ,which is presented the relationship between cytokine polymorphism and neonatal sepsis) ” section according to your suggestion (Pages 10-14, 15-16, Lines 191-196, 218-256).

REVIEWER COMMENTS:

4. The limitations of this study could be more comprehensively described in the discussion section. Currently, only heterogeneity and sample size are discussed. Whether there are other biases that need to be explored.

AUTHOR RESPONSE:

We think this is an excellent suggestion. We have added these instructions in the “4. Discussion” section according to your suggestion (Pages 45-46, Lines 820-828).

REVIEWER COMMENTS:

5. The conclusion is a bit absolute. It is suggested to use words like “indicate”, “imply” instead of “confirm”, as this study cannot completely “confirm” the associations between certain genes and disease.

AUTHOR RESPONSE:

As suggested by the reviewer, we have revised these instructions in the “4. Discussion” section according to your suggestion (Pages 44-45, Lines 795-806).

REVIEWER COMMENTS:

6. The format of references could be unified.

AUTHOR RESPONSE:

We think this is an excellent suggestion. We have revised these instructions in the “References” section according to magazine requirements (Pages 47-56, Lines 847-1050).

REVIEWER COMMENTS:

7. The excessive number of pictures is very inconvenient for readers.

AUTHOR RESPONSE:

Thank you for your constructive suggestion. Due to the inclusion of 5 cytokines in our study, each cytokine was analyzed separately from 6 gene models. Therefore, we obtained a large number of images through meta-analysis. To demonstrate the conciseness and clarity of the manuscript, based on your suggestion, the results of each inspection have been included in the table, and some images have been removed. And attach the images for those who need them to review.

REVIEWER COMMENTS:

8. The language context in the manuscript is less coherent and the writing is more mechanical, so it is recommended to have it revised by a professional with a foundation in English as a mother tongue.

AUTHOR RESPONSE:

Thanks for your suggestion. There are indeed shortcomings in our language. However, we do invite a friend of ours who is a native English speaker from the USA to help polish our article. These changes will not influence the content and framework of the paper. Here we did not list the changes but marked them in red in the revised paper. We appreciate for Academic Editor and Reviewers’ warm work earnestly and hope that the correction will meet with approval.

Reviewer #2

REVIEWER COMMENTS:

1.General. English grammar needs to be improved, though it is comprehensible but not upto the mark.

AUTHOR RESPONSE:

Thanks for your suggestion. There are indeed shortcomings in our language. However, we do invite a friend of ours who is a native English speaker from the USA to help polish our article. These changes will not influence the content and framework of the paper. Here we did not list the changes but marked them in red in the revised paper. We appreciate for Academic Editor and Reviewers’ warm work earnestly and hope that the correction will meet with approval.

REVIEWER COMMENTS:

2.Title. The title correctly reflects the aim and scope of the research.

AUTHOR RESPONSE:

Thank you very much for the strong support to our work.

REVIEWER COMMENTS:

3. ABSTRACT

The abstract follows a logical format, outlining the background, objectives, methods, results, and conclusions. Searching multiple databases increases the likelihood of capturing relevant literature. Appropriate statistical tests are used to assess heterogeneity and publication bias. The abstract details the significant associations identified for different cytokines and genetic models.

The abstract acknowledges the limitations of small sample size in some studies.

i. While the abstract states associations between specific polymorphisms and neonatal sepsis, it could benefit from mentioning the direction of the effect (e.g., increased/decreased risk).

ii. Quantifying the strength of evidence using measures like P values and confidence intervals would strengthen the abstract.

iii. Briefly mentioning the potential clinical significance of the findings (e.g., for diagnosis or personalized medicine) could enhance the abstract's impact.

Collectively, this is a well-written abstract that effectively summarizes the main findings of the meta-analysis.

AUTHOR RESPONSE:

Firstly, thank you very much for the strong support of our work. Based on your suggestion, we have revised the content of the abstract in the “Abstract” section (Page 2, Lines 26-48), and added content on the effects aspect. Quantitative indicators such as P-value and confidence interval were used. Finally, the potential clinical significance of the research results was briefly mentioned. Due to the word limit of the abstract in PLOS ONE magazine, we have streamlined the overview and provided detailed information in the main text. Thank you again sincerely for your support in our work.

REVIEWER COMMENTS:

4. 1. INTRODUCTION

The introduction section of the meta-analysis clearly states the importance of neonatal sepsis and its connection to inflammation and also highlights the role of cytokines in immune response and potential connection to sepsis susceptibility. It also mentions the inconsistency of previous research on cytokine polymorphisms and neonatal sepsis and also clearly describes the aim of the current study

i. Though the introduction section is good but very short and broad. It could be improved by broad-to-narrow manner, to a more specific statement about the study's hypothesis or key research question.

ii. Repetition. Sentences 59-60 and 63 could be combined to avoid repetition.

iii. In the reviewer opinion, the meta-analysis needs to be supported by more and recent reference (if available). References are only used once in the introduction, making it difficult to assess the strength of the claims made. Consider incorporating more specific citations and evidence to support the statements.

iv. Information on the specific polymorphisms explored in the study is missing. Mentioning which cytokine genes and polymorphisms will be investigated would provide better context.

v. The introduction primarily focuses on genetic susceptibility and doesn't mention other potential factors influencing the association between cytokines and neonatal sepsis (e.g., environmental factors, maternal health). Briefly acknowledging these complexities would provide a more nuanced picture of the research topic.

vi. Conclude the introduction section with a clear statement of the study's objectives and its potential contribution to the field.

AUTHOR RESPONSE:

Firstly, thank you for your recognition of our work. Based on your suggestion, we have supplemented the introduction section in the “1.Introduction” section (Pages 3-4, Lines 50-89), and merged duplicate content, as well as added citations to the references. A brief mention of which cytokines and polymorphisms provide a better background for our research. And other potential factors affecting the relationship between cytokines and neonatal sepsis were mentioned. Finally, we mentioned the potential contribution of our research to this field. Thank you again sincerely for your support in our work.

REVIEWER COMMENTS:

5. 2. Materials and methods

2.1Document retrieval

The terms provided seem good, but justification for their inclusion (e.g., pilot searches, previous reviews) would support the rationale of the search strategy.

Line 72. Please rephrase the sentence to. To ensure a comprehensive search for relevant articles, we combined medically relevant keywords with broader terms. Additionally, when encountering duplicate publications, we selected the one with the most complete data for inclusion in the study.

For the above statement, the reviewer would like to know “How is the "most complete data" chosen among repeated studies?”. Explain in one or two lines.

AUTHOR RESPONSE:

Firstly, thank you for your recognition of our work. Based on your suggestion, we have added reasons for including these terms in the “2.1 Document retrieval” section (Pages 4-5, Lines 92-95, 97-102),and modified the sentence in line 72 of the original unrevised manuscript, addressing your question of “how to select the” most complete data “in repeated studies” in the “2.1 Document retrieval” section (Page 5, Lines 99-102) .

REVIEWER COMMENTS:

6. 2.2Inclusion and exclusion criteria

The meta-analysis carefully selected relevant studies by focusing on specific designs, prioritizing newborn populations, ensuring data quality, and excluding redundant or irrelevant research. Excellent.

Limiting the study to Chinese and English might exclude valuable research conducted in other languages. Consider justifying this choice or exploring translation options.

Like Figure 1, almost all other figures are not visible and readable though they seem correct. The resolution could be resolved for clarity, for all figures.

AUTHOR RESPONSE:

Firstly, thank you for your recognition of our work. When searching for relevant literature, based on the Chinese and English search terms set by previous similar literature attempts, the retrieved literature in other languages was not found to be relevant to the research in this article after translation. Based on your suggestion, the sentence "and the literature types were limited to Chinese and English" has been removed from the manuscript. And due to the formatting requirements of PLOS ONE magazine, images are required to be uploaded separately, but the order and name of the images are required to be listed in the manuscript. And because the inclusion of 5 cytokines in our study, each cytokine was analyzed separately from 6 gene models. Therefore, we obtained a large number of images through meta-analysis. To demonstrate the conciseness and clarity of the manuscript, based on your suggestion, the results of each inspection have been included in the table, and some images have been removed. And attach the images for those who need them to review.

REVIEWER COMMENTS:

7. 2.3Literature screening

Please remove the follow

---

## [Editor Report · Decision Letter 1]

24 Mar 2024

A meta-analysis of the association between inflammatory cytokine polymorphism and neonatal sepsis

PONE-D-23-36527R1

Dear Dr. Hao,

We’re pleased to inform you that your manuscript has been judged scientifically suitable for publication and will be formally accepted for publication once it meets all outstanding technical requirements.

Kind regards,

Mohammad Safiqul Islam, Ph.D

Academic Editor

PLOS ONE

Additional Editor Comments (optional):

The authors revised the manuscript according to the suggestion of reviewers and academic editors. I recommend for acceptance in the current form of the manuscript.
---

## [Editor Report · Acceptance letter]

5 Apr 2024

PONE-D-23-36527R1 

PLOS ONE

Dear Dr. Hao, 

I'm pleased to inform you that your manuscript has been deemed suitable for publication in PLOS ONE. Congratulations! Your manuscript is now being handed over to our production team.

Kind regards, 

on behalf of

Dr. Mohammad Safiqul Islam 

Academic Editor

PLOS ONE